# A biochemical mechanism for time-encoding memory formation within individual synapses of Purkinje cells

**Ayush Mandwal** [1]*, **Javier G. Orlandi**[1], **Christoph Simon**[2,3], **Jörn Davidsen**[1,3]*

**1** Complexity Science Group, Department of Physics and Astronomy, University of Calgary, Calgary, Alberta, Canada, **2** Department of Physics and Astronomy, University of Calgary, Calgary, Alberta, Canada, **3** Hotchkiss Brain Institute, University of Calgary, Calgary, Alberta, Canada

* ayush.mandwal@ucalgary.ca (AM); davidsen@phas.ucalgary.ca (JD)

**Data Availability Statement:** All relevant data are within the paper and its Supporting information files.

## Abstract

Within the classical eye-blink conditioning, Purkinje cells within the cerebellum are known to suppress their tonic firing rates for a well defined time period in response to the conditional stimulus after training. The temporal profile of the drop in tonic firing rate, i.e., the onset and the duration, depend upon the time interval between the onsets of the conditional and unconditional training stimuli. Direct stimulation of parallel fibers and climbing fiber by electrodes was found to be sufficient to reproduce the same characteristic drop in the firing rate of the Purkinje cell. In addition, the specific metabotropic glutamate-based receptor type 7 (mGluR$_7$) was found responsible for the initiation of the response, suggesting an intrinsic mechanism within the Purkinje cell for the temporal learning. In an attempt to look for a mechanism for time-encoding memory formation within individual Purkinje cells, we propose a biochemical mechanism based on recent experimental findings. The proposed mechanism tries to answer key aspects of the "Coding problem" of Neuroscience by focusing on the Purkinje cell's ability to encode time intervals through training. According to the proposed mechanism, the time memory is encoded within the dynamics of a set of proteins—mGluR$_7$, G-protein, G-protein coupled Inward Rectifier Potassium ion channel, Protein Kinase A, Protein Phosphatase 1 and other associated biomolecules—which self-organize themselves into a protein complex. The intrinsic dynamics of these protein complexes can differ and thus can encode different time durations. Based on their amount and their collective dynamics within individual synapses, the Purkinje cell is able to suppress its own tonic firing rate for a specific time interval. The time memory is encoded within the effective dynamics of the biochemical reactions and altering these dynamics means storing a different time memory. The proposed mechanism is verified by both a minimal and a more comprehensive mathematical model of the conditional response behavior of the Purkinje cell and corresponding dynamical simulations of the involved biomolecules, yielding testable experimental predictions.

**Funding:** This work was financially supported by the Natural Sciences and Engineering Research Council of Canada through a Discovery Grant to JD and the Eyes High Initiative of the University of Calgary (JD,JO). There was no additional external funding received for this study. The funders had no role in study design, data collection and analysis, decision to publish, or preparation of the manuscript.

**Competing interests:** The authors have declared that no competing interests exist.

## Introduction

How do we store memories in our brain? How do we retrieve and edit them when required? Recent experimental findings have shed some light onto these fundamental questions. Experiments have shown that memories are held within specific neuronal populations [1–3]. Such populations, referred as *memory engram cells* [4, 5] store memory either by forming or eliminating synapses [6, 7] or by altering synaptic strengths between neurons [8, 9] within the population. These forms of learning and memory formation fall under the widely accepted Hebbian learning paradigm [10]. However, the individual contribution of each synapse to the engrams, and how changes in synaptic strength affects memories, remain poorly understood. The problem of information encoding was raised by C.R. Gallistel [11] and termed as the "Coding Question", one of the fundamental open questions in Neuroscience today. Recent experiments on Purkinje cells, one of the major neuronal populations in the Cerebellum and essential for motor coordination, have shed some light on the Coding Problem. Those experimental results have illustrated that the memory of time interval duration can be encoded within individual Purkinje cells, and does not require a whole neuronal population [12, 13]. In addition, the stored time memory can be accessed and changed anytime. This result has also challenged the prevailing doctrine of Hebbian learning by showing that traditional changes of synaptic strength alone cannot explain the Purkinje cell response after learning [14].

Purkinje cells can learn to encode a specific time memory through Classical or Pavlovian conditioning. This kind of associative learning can occur when a biologically potent stimulus, such as food, is paired with a neutral stimulus, such as a metronome, that precedes it. Depending upon the response the potent stimulus elicits, e.g., saliva flow, and the exact protocol followed, Classical Conditioning can be categorized into various kinds. One of them being classical motor conditioning, such as the eye blink conditioning, where a neutral conditional stimulus (CS) in the form of a light or a sound can trigger an eye blink response before the onset of an unconditional stimulus (US) that elicits a blink reflex response [15, 16]. In other words, CS triggers a response that predicts the time of arrival of the US. Such a conditional response appears after training with repeated paired presentation of two stimuli, where a CS is followed by an US after a fixed time interval "T", called the interstimulus time interval (ISI) [17]. At the cellular level, the eye blink response is causally related to a suppression of the tonic firing of individual Purkinje cells, which have projections onto cerebellar nuclei which in turn regulate the activity of ocular muscles [17, 18]. Because of such causal connection, the suppression of the firing rate of the Purkinje cell is termed as the conditional response of the Purkinje cell.

Previous mechanistic explanations considered Long-term Depression (LTD) of selective synapses between parallel fibers and Purkinje cells (pf-PC) as the main mechanism behind the conditional response in the Purkinje cell [19]. Based on the widely accepted Marr-Albus model of the cerebellum [20, 21], this suggests that the time memory of the response is encoded within the network dynamics of Granule cell neurons and inhibitory interneurons, found within the molecular layer of the Cerebellum between Mossy fibers and Purkinje cells. However, recent experiments on ferrets were able to identify the source of the conditional response at the level of *individual* Purkinje cells by showing that the direct stimulation of parallel fibers and climbing fibers using electrodes was sufficient for Purkinje cells to learn the specific time interval duration [12]. These experiments also showed that a glutamate-based metabotropic receptor type 7 (mGluR7) initiates the conditional response [13] by opening G-protein coupled Inward Rectifier Potassium (GIRK) ion channels [22]. This implies that there exists a specific biochemical mechanism within the Purkinje cell that can encode and store temporal information. However, immunohistochemistry results for the expression of the

mGluR$_7$ receptor on the Purkinje cell's synapses or dendritic spines have been highly controversial. Although several early studies showed mGluR$_7$ expression in Purkinje cells [23, 24], posterior with proper controls and highly specific immunostaining concluded that Purkinje cells lack mGluR$_7$a and mGluR$_7$b-like immunoreactivities on dendritic spines or cell bodies completely [25]. Instead these authors proposed that mGluR$_7$b might be expressed on the Purkinje cell axon's terminals. Considering moderate or fairly strong mGluR$_7$ mRNA expression levels in Purkinje cells [26, 27], one expects a significantly wider expression of mGluR$_7$ receptors on the Purkinje cell. Although the study in [25] applied a commonly applicable immunostaining approach, it is possible that it was not sufficiently sensitive to detect low or moderate expression of mGluR$_7$ receptors on the remaining parts of the Purkinje cells as proper immunostaining depends on both specificity and sensitivity of the antibody-antigen pair [28, 29].

In summary, it has been traditionally believed that memory storage in the cerebellum requires neuronal assemblies. The recent findings suggest instead that temporal signatures can be encoded within a single Purkinje cell. Yet, the specific mechanism remains poorly understood. Here, we propose a biochemical description, based on past experimental findings, that is able to explain time memory formation, consolidation and access.

## Materials and methods

### Model conceptualization: Proposed biochemical mechanism

As mentioned above, the conditional response at the level of an individual Purkinje cell appears after several repetitions of two stimuli: A CS from the parallel fibers followed by an US from the climbing fiber after a fixed ISI. A sufficient condition for the learning process to be called completed is that a CS without an applied US can initiate the conditional response—the suppression of the tonic firing rate—within the given ISI.

We propose that the conditional response arises due to interactions between four main proteins: metabotropic glutamate based receptor (mGluR$_7$), Protein Kinase A (PKA), Protein Phosphatase 1 (PP1) and G-protein, which regulate the gating dynamics of the G-protein inward rectifier potassium (GIRK) ion channel. G-Protein Coupled Receptors such as the mGluR$_7$ receptor have been known to form protein complexes with GIRK ion channels [30–32]. The protein complex can include Phosphatase and Kinase proteins such as PP1 which can be active all the time, while the activity of PKA can change depending on the cAMP concentration [33, 34]. As PKA and PP1 activities have opposite roles, one of the two proteins will typically dominate and decide the de/phosphorylation state of the target protein [33, 35]. For instance, PKA dominates over PP1 upon increase in cAMP concentration, [cAMP], and causes phosphorylation of the target protein [33, 35]. Otherwise PP1 dominates and causes dephosphorylation of the target protein [35]. Such dual role of Kinase and Phosphatase have been observed to be facilitated by A-Kinase Anchoring Proteins (AKAPs) [31–33]. In addition, AKAP proteins can also harbour Acetyl-Cyclase (AC) proteins, which can associate with G-protein coupled receptors to regulate the receptor mediated ion channel dynamics [31, 33]. Altogether, we propose that the mGluR$_7$ receptor, G-protein and GIRK ion channel form a stable protein complex along with proteins like AC, PKA and PP1, which are associated via an AKAP scaffold protein close to the receptor.

Below, we provide more detail on the conditional response of the Purkinje cell and the associated detailed biochemistry we propose. We separate our discussion into three parts: during training, after training and training with different ISIs. The first part focuses on two questions: What makes a Purkinje cell learn a conditional response, and how does the cell learn a conditional response of a specific duration? The later two parts describe the most crucial aspects of

the conditional response, i.e., its formation after training, along with other features of the conditional response, which were experimentally observed.

**During training: Learning process.** What makes a Purkinje cell learn a conditional response? The activation of the conditional response was found to be initiated by the activation of mGluR$_7$ receptors [13]. Although the body of literature regarding mGluR$_7$ in the cerebellum is limited and despite the aforementioned controversy regarding the expression of the mGluR$_7$ receptor on the Purkinje cell's synapses or dendritic spines, there is significant evidence that Purkinje cells do express mGluR$_7$ receptors. Most importantly, the effects of 6-(4-Methoxy-phenyl)-5-methyl-3-(4-pyridinyl)isoxazolo[4,5-c]pyridin-4(5H)-one hydrochloride (MMPIP) as an mGluR$_7$ selective antagonist replicate the results of mGluR$_7$ knockouts [36], while MMPIP also has an effect on blocking the conditional response in Purkinje cells [13]. Thus, we start from the assumption that Purkinje cells express mGluR$_7$ on the Purkinje cells' synapses and that the mGluR$_7$ receptors indeed activate the conditional response behaviour in the Purkinje cell. Yet no conditional response was observed before training [12]. We propose that during training mGluR$_7$ receptors are being transported from the perisynaptic zone to the postsynaptic zone of the synapse. Alternative hypotheses such as (1) the absence of GIRK ion channels at the synapse and (2) low expression of G$_{i/o}$ type G-proteins at the synapse can be ruled out. Immunohistochemistry analysis showed the presence of GIRK subtypes GIRK2/3 ion channels on the synapses of Purkinje cells—which are innervated by parallel fibers [37]. If (2) were true and the G-protein expression would change during training, this would affect not only the conditional response profile but also various other physiological properties of the Purkinje cell. This is because different types of G-proteins play crucial roles in signal transductions and determine various physiological properties of the cell [38]. Since no change in the tonic firing rate has been observed before and after conditional training [12], we believe that other physiological properties of the cell may also remain unaltered. Thus, the translocation of mGluR$_7$ receptors to the synapse is the most likely result of the training and we assume that the amount of other proteins such as GIRK ion channels and G-protein is constant for all different durations of conditional training.

How does the Purkinje cell learn a conditional response of a specific duration? Purkinje cells memorize a specific duration "T" after training with repeated paired presentation of two stimuli, where a CS is followed by an US after a fixed time interval "T" i.e., the Interstimulus Interval (ISI) [17]. As mentioned earlier, we propose that the learning of the conditional response is associated with trafficking of mGluR$_7$ receptors from perisynaptic to postsynaptic locations at the Purkinje cell's synapses. Specifically, we propose that such trafficking of receptors occurs via Clathrin-mediated Endocytosis (CME) mediated by the activation of Protein Kinase C (PKC) [39]. The PKC activation occurs in the presence of two stimuli: the first stimulus must come from the parallel fiber, which activates mGluR$_1$ receptors, while the second stimulus from the climbing fiber raises the Ca$^{+2}$ ion concentration [40, 41]. Both stimuli are necessary and, in particular, the presence of only one of the two stimuli is not sufficient for either PKC activation or Purkinje cell to learn the conditional response [17, 40]. Consequently, we also propose that mGluR$_1$ receptors are essential for learning of a conditional response of a specific duration. There could be other biochemicals involved in the translocation of mGluR$_7$ receptors as Purkinje cells cannot be trained for ISI durations shorter than 100msec [42]. Currently, we cannot make any suggestion for proteins, which might be involved in addition to PKC during conditional learning.

To ensure storage of a specific time duration memory, such translocation processes must stop after some time. This can happen by inhibiting PKC activation via the activation of GIRK ion channels. Indeed, activation of GIRK ion channels causing a drop in tonic firing rate during training has been observed in experiments [43, 44]. Therefore, we propose that as training

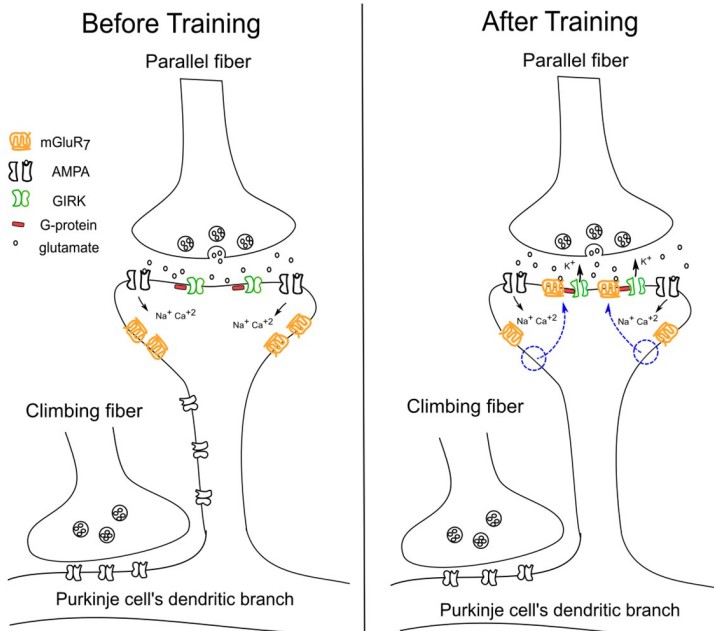

**Fig 1. mGluR$_7$ receptor distribution before and after conditional training in the Purkinje cell.** Before training, mGluR$_7$ receptors are localised at perisynaptic areas of the synapses. After training, as pointed out by the blue arrows, these receptors localised themselves at the postsynaptic area of the synapse via CME.

progresses the intracellular Ca$^{+2}$ ion concentration decreases to a level that is no longer sufficient to activate PKC, which prevents further translocation of mGluR$_7$ receptors to the synapse and so a steady state will be reached. When a steady state has been reached, then we can say that the Purkinje cell has learned the conditional response of duration "T" as shown in (Fig 1). This learning mechanism also suggests that the training period needs to increase with the duration "T" as observed experimentally [12]. As the net amount of the receptors translocated during training depends upon its net duration, longer training means more transportation of the receptor to the synapse. We will explain below how a higher amount of receptors can produce a longer duration conditional response.

**After training.** The conditional response with a duration of hundreds of milliseconds can be initiated by a CS of as little as 20 milliseconds duration [12]. This means that just the activation of the mGluR$_7$ receptors by CS is enough to initiate the conditional response, which is only possible if the receptor remains active even after CS is over. Furthermore, in order to explain the fast dynamics of the conditional response initiation observed in the experiment [12] we propose that the mGluR$_7$ receptor forms a protein complex with the G-protein of G$_{i/o}$ type, which is located in direct vicinity to a GIRK ion channel as facilitated by a Regulator of G-protein signaling protein 8 (RGS8) [45, 46]. RGS8 proteins are expressed in dendritic spines of the Purkinje cell [47] and they have the special property of accelerating both activation and deactivation of the G-protein causing fast opening and closing of GIRK ion channels [46, 48].

If the mGluR$_7$ receptors can remain active even after CS is over, there needs to be mechanisms by which they can return to an inactive state. To explain this, there are two additional important properties of the conditional response, which we must consider: 1) The conditional response is lost after repetitive CS [12], and 2) the conditional response is independent of CS duration. Dephosphorylation of the mGluR$_7$ receptor by Protein Phosphatase 1 (PP1), which causes their rapid internalization [49], can explain both these properties of the conditional

response. Rapid internalization of any receptor is initiated by the binding of a protein called Arrestin protein, which prevents the receptor to transmit any signal further [41]. Because of rapid internalization, retraining of the Purkinje cell with the same or a different ISI will be faster as many receptors are close to the synapse. This rapid relearning phenomenon is called "Saving" and it takes only a few minutes to recall the old memory of the conditional response by the Purkinje cell [17]. To prevent dephosphorylation and rapid internalization of $mGluR_7$ receptors, Calmodulin can stimulate Acetyl cyclase (AC) [50] to produce cAMP molecules and increase PKA activity. In addition, Calmodulin can also activate PDE enzymes [51] which will limit the PKA activity. It is also known that PKA can phosphorylate $mGluR_7$ receptors [52]. Thus, phosphorylation of the receptor depends on the competition between PKA and PP1 activity as in [33], where PKA dominates over PP1's constant activity and causes net phosphorylation of the $mGluR_7$ receptor. Thus, the phosphorylation of receptors by PKA helps in the retention of the memory for a long time. As PKA and PP1 are essential for the conditional response, we propose that they bind to the receptor via a AKAP protein [53].

In short, the underlying biochemical mechanism of the conditional response can be described as follows. The release of glutamate during CS activates $mGluR_7$ receptors on the Purkinje cell synapses [step 1 of (Fig 2)], which in turn activates G-proteins [step 2 of (Fig 2)]. Each unit of G-protein splits into a $G_\alpha$ subunit and a $G_{\beta\gamma}$ subunit. One unit of $G_\alpha$ subunit binds to an AC enzyme to block the production of cAMP molecules. This in turn deactivates PKA as Phosphodiesterase enzyme (PDE) hydrolyses the remaining cAMP molecules [41] [step 3 of (Fig 2)]. At the same time the $G_{\beta\gamma}$ subunit binds to the GIRK ion channel, which becomes fully active upon binding of four $G_{\beta\gamma}$ subunits [54]. As PKA activity decreases, PP1 activity causes dephosphorylation of $mGluR_7$ receptors [step 4 of (Fig 2)] and initiates their rapid internalization. However, rapid internalization of a receptor is still a slow process compared to the conditional response as it involves many protein interactions and, hence, the receptor is not immediately displaced from the synapse after dephosphorylation. However, after dephosphorylation, Arrestin protein blocks the active site of the $mGluR_7$ receptor to prevent reactivation of the G-protein [55] as well as decouples the receptor from the protein complex [step 5 of (Fig 2)]. After receptor dephosphorylation, the active G-protein is deactivated by the RGS8 protein [step 6 of (Fig 2)]. As G-protein activity reduces, GIRK ion channels also shut down. In the absence of active G-protein, PKA activity begins to rise again [step 7 of (Fig 2)] due to rise in activity of AC enzymes in the presence of Calmodulin. Active PKA phosphorylates $mGluR_7$ receptors [step 8 of (Fig 2)] to prevent their internalization and the uncoupled phosphorylated receptor recouples back to the protein complex to prepare the Purkinje cell for another conditional response. It is likely that the reactivation of PKA takes some time, which might explain why CS cannot initiate another conditional response while CS is still on.

The rate at which GIRK ion channels open and close depends upon the rate at which intermediate reactions occur. In other words, the time memory of the training is stored within the effective dynamics arising from these reactions. In a complete cycle of GIRK ion channel activation and deactivation, altering only the effective dynamics for both activation and deactivation processes is sufficient to store a different time memory of the conditional response.

**Training with different ISI duration and time-encoding protein complexes.** Training with a different ISI duration means storage of a different time memory. There are two additional questions we need to answer in order to get a complete understanding of time memory storage in biochemical reactions: 1) How do these biochemical reactions get tuned so finely to store a specific time duration memory? 2) Which dynamical parameters of the proposed biochemical mechanism are most likely to get affected by choosing a different ISI for the training?

The reason behind 1) is that there are several GIRK ion channels present at the synapse. Each GIRK ion channel requires four units of $G_{\beta\gamma}$ subunits to open completely [54]. This

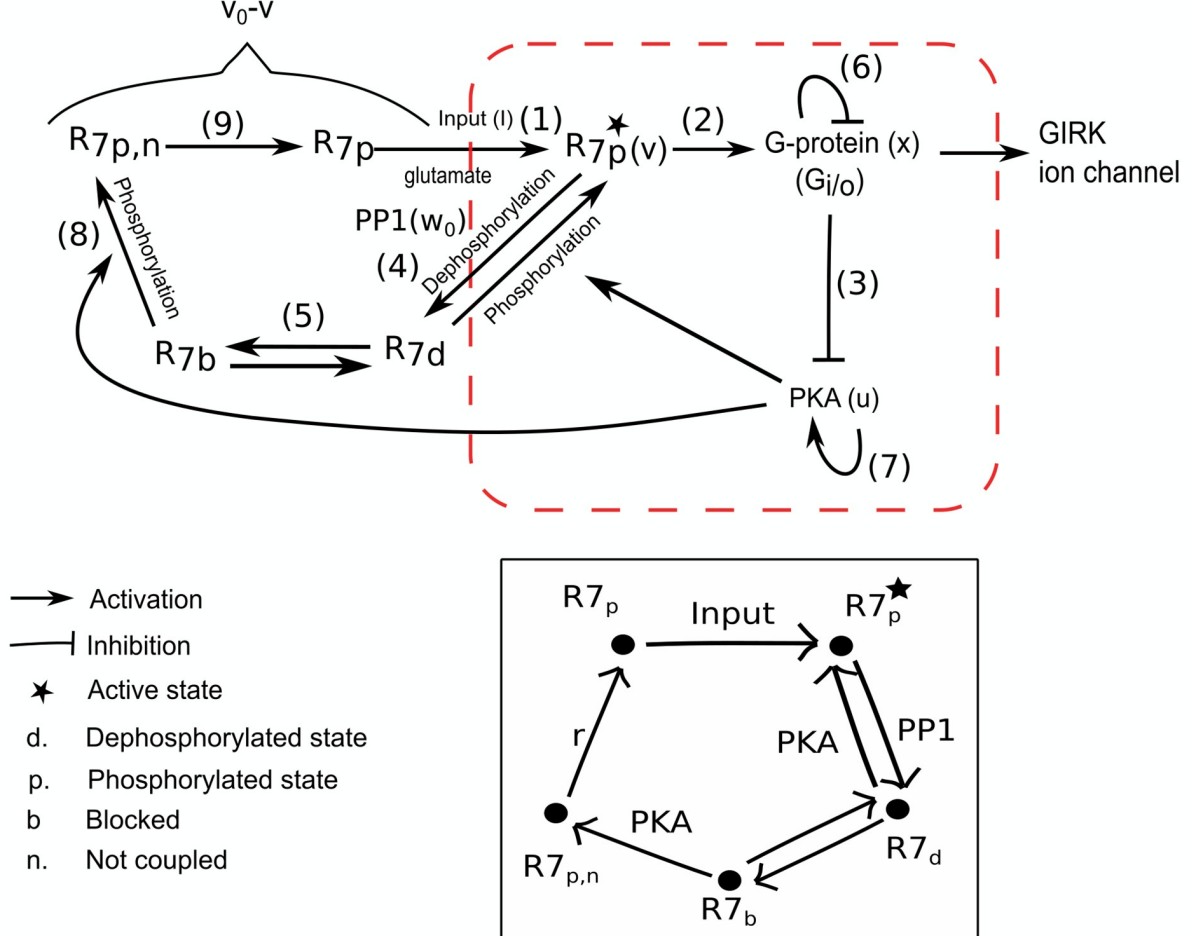

**Fig 2. Interactions between different biochemicals involved in our proposed mechanism of the conditional response in the Purkinje cell.**
Abbreviations: R7—mGLuR$_7$ receptor, PKA—Protein Kinase A, PP1—Protein Phosphatase 1, GIRK—G-protein coupled Inward Rectifier Potassium ion channel. The numbers on the top of the arrows highlight the order in which the different reactions occur during the conditional response. Conditional response initiates with the release of glutamate from parallel fibers denoted by I as input in (1), which activates mGluR$_7$ receptors. In (2), active receptors activate G-proteins, which deactivate PKA through (3). As PKA activity reduces, PP1 activity becomes dominant (4) causing dephosphorylation of the receptor. The dephosphorylated receptor will be blocked by Arrestin protein to prevent further signaling transduction and also decouples the receptor from the protein complex (5). As receptor activity reduces, RGS8 reduces G-protein activity (6), which allows PKA activity to rise again (7). Active PKA will phosphorylate dephosphorylated receptors (8) to prevent their rapid internalization and the uncoupled phosphorylated receptor will couple back to the protein complex (9) for another conditional response. The red box identifies the three variables and their interactions used in the minimal mathematical model to capture the conditional response behaviour. Here, $u$, $v$ and $x$ represent the activities of PKA, mGluR$_7$ receptor and G-protein, respectively. These protein interactions occur in each individual TEC unit present at the synapse. The remaining states of the mGluR$_7$ receptor are collectively denoted as $v_0 - v$, where $v_0$ is the maximum possible activity of the mGluR$_7$ receptor. Below, in the black box we highlight the pictorial description of the 5-state model of mGluR$_7$ receptors, which is explicitly used in the comprehensive mathematical model.

means that each GIRK ion channel forms a protein complex with four units of each of G-proteins, receptors and RGS8 proteins along with PKA and PP1 proteins together with their anchoring proteins. As each of these protein complexes has their own intrinsic dynamics, which regulate how fast the GIRK ion channel opens and closes upon stimulation, we can call each of these protein complexes "Time-Encoding protein Complexes" (TEC). Within each TEC, the rate of G-protein activation by the receptor and the rate of binding of G-protein subunits to the GIRK ion channel decide the overall rate of opening of GIRK ion channels i.e., the

onset of the conditional response. After the onset of the conditional response the rates of PKA deactivation, dephosphorylation of the receptor by PP1 and the deactivation of G-protein by RGS8 decide the overall duration of the conditional response since at the end of these biochemical reactions the GIRK ion channel begins to close. Thus, each TEC encodes the time information of the conditional response completely in terms of the effective dynamics of different biochemical interactions and stores this time memory by forming a protein complex. Formation of a protein complex as TEC ensures strong consolidation of memory with less chances of errors in the information storage. If the rates were to be changed so would the memory as well. The rates can be affected by the translocation of extra mGluR$_7$ receptors to the synapse during conditional training. These extra mGluR$_7$ receptors can form clusters with receptors—which are part of a TEC—with the help of a scaffold protein, Protein Interacting with C Kinase—1 (PICK1) [56]. Such cluster formation can affect TEC's intrinsic dynamical properties by influencing the protein interaction of the mGluR$_7$ receptor with the G-protein facilitated by RGS8. As a result, RGS8's ability to accelerate the dynamics of the conditional response might be affected, which results in a delayed onset of the conditional response. Such clustering of receptors can also affect the concentration of PDE proteins anchored close to the receptor via the AKAP protein, thus affecting the rate at which PKA deactivates and hence the time duration of the conditional response. To summarize, we propose that at individual synapses the interaction of extra mGluR$_7$ receptors with TECs can affect the dynamics of TECs and collectively these varied TEC units help to produce the conditional response of any specific time duration in the Purkinje cell.

From the above description, it follows that in principle a single synapse can completely contain a time duration memory, which can be altered through retraining. However, a single synapse probably will not be sufficient to suppress the tonic firing rate of the whole Purkinje cell. This is because the spontaneous tonic firing rate of the Purkinje cell [57] appears to be due to voltage-dependent resurgent Na$^+$ ion channels, which are distributed over the *entire* somata and dendritic regions of the cell [58, 59]. Activation of GIRK ion channels by CS can hyperpolarize the membrane at a synaptic region and deactivate resurgent Na$^+$ ion channels near this synaptic region. Thus, a finite fraction of the total pf-PC synapses distributed over the dendrites can produce a suppression in tonic firing rate of the Purkinje cell for a specific duration and the corresponding memory is encoded at the respective synapses.

## Model implementation

Here, we provide both a minimal and a comprehensive mathematical model implementation of our proposed biochemical mechanism in the context of an existing Purkinje cell model.

**Purkinje cell model.** To model the conditional response behavior of the Purkinje cell after training, we start with an established dynamical model of the Purkinje cell [60] as summarized by (Eqs 1–5). Specifically, it aims to model the dynamics of the Purkinje cell by incorporating many properties of the Purkinje cell within a realistic biophysical framework. In contrast to the original formulation [60], (Eqs 1–5) already incorporate the features specific to our situation: In (Eq 1), the input current term $I_i$, which originally signified an external electrical stimulus, now signifies the intrinsic current causing the tonic firing of the Purkinje cell [61, 62]. Moreover, before training, GIRK ion channels cannot be opened because mGluR$_7$ receptors are not present at the synapse, while after training, mGluR$_7$ receptors are present at the synapse to open GIRK ion channels. Therefore, we added the influence of the GIRK ion channel in (Eq 2), which only becomes relevant *after* training. Here, $g_{GIRK}$ is the net conductance of GIRK ion channels per unit area, $h_{GIRK}$ is the gating parameter and $V_{GIRK}$ is the voltage dependence of the GIRK ion channel obtained from the I-V characterstics curve of the ion channel

[63]. To model the conditional response behavior of the Purkinje cell after training, we capture the dynamics of our proposed biochemical mechanism using the gating parameter $h_{GIRK}$. As a GIRK ion channel binds 4 units of G-protein subunits, i.e., $G_{\beta\gamma}$, we have used the exponent 4 for the $h_{GIRK}$. This is based on the assumption that the dynamics of each unit of $G_{\beta\gamma}$ is independent of the others. We will discuss later the (very limited) effect of relaxing this assumption, including the case when they are all strongly dependent on one another corresponding to an exponent of 1.

Except for $g_{GIRK}$, all values of the model are taken from [60]. As far as we know, there is no literature on the specific $g_{GIRK}$ values. As a result, we chose a value of $g_{GIRK}$ that matches the experimentally observed conditional response profiles. All parameter values of the Purkinje cell model including $g_{GIRK}$ are summarized in S1 Text

Somatic voltage equation:

$$C_s \frac{dV_s}{dt} = \frac{(V_d - V_s)}{R} - g_{Na}m_\infty h(V_s - E_{Na}) - g_{Ks}(1-h)(V_s - E_K)$$
$$-g_{leak}(V_s - E_{leak}) - g_{I_H}I_h(V_s - E_{I_h}) + I_i \tag{1}$$

Dendritic voltage equation:

$$C_d \frac{dV_d}{dt} = \frac{(V_s - V_d)}{R} - g_{leak}(V_d - E_{leak}) - g_{Kd(slow)}n_d(V_d - E_K)$$
$$-g_{GIRK}h_{GIRK}^4 V_{GIRK}(V_d) \tag{2}$$
$$V_{GIRK}(V_d) = -0.02(1.3V_d + 50.0)/(1.0 + exp((V_d + 40)/10.0))$$

Na$^+$ activation equation:

$$m_\infty = \frac{1}{1 + exp[-(V - V_{1/2})/k]}, \quad V_{1/2} = -40.0mV, \quad k = 3.0mV$$

$$\frac{dh}{dt} = \frac{h_\infty - h}{\tau_h} = \frac{1}{1 + exp[-(V - V_{1/2})/k]}, \quad V_{1/2} = -40.0mV, \quad k = -3.0mV \tag{3}$$

$$\tau_h(V) = \frac{295.4}{4(V + 50)^2 + 400} + 0.012$$

Hyperpolarizing activated cation current ($I_h$):

$$\frac{dI_h}{dt} = \frac{I_{h\infty} - I_h}{\tau_{I_h}} = \frac{1}{1 + exp[-(V - V_{1/2})/k]}, \quad V_{1/2} = -80.0mV, \quad k = -3.0mV,$$
$$\tau_{I_h} = 100ms \tag{4}$$

Slow K$^+$ activation equation:

$$\frac{dn_d}{dt} = \frac{n_{d\infty} - n_d}{\tau_{nd}} = \frac{1}{1 + exp[-(V - V_{1/2})/k]} \quad V_{1/2} = -35.0mV, \quad k = 3.0mV$$
$$\tau_{n_d} = 15ms \tag{5}$$

**Minimal mathematical model of proposed biochemical mechanism.** We first start with a minimal mathematical model for the dynamics of the gating parameter $h_{GIRK}$, which will

allow us later to clearly establish the robustness and generality of the proposed mechanism since it is amenable to a phase space and bifurcation analysis.

Gating of the GIRK ion channel depends upon the availability of Phosphatidylinositol 4,5-bisphosphate ($PIP_2$) molecules [54]. These molecules have a low affinity for GIRK ion channels but bind efficiently after binding of a $G_{\beta\gamma}$ subunit to a GIRK ion channel. The amount of $PIP_2$ on the synaptic membrane is low but it is replenished by various biochemical processes to maintain its concentration fairly constant upon consumption or degradation [64]. Therefore, the amount of active $G_{\beta\gamma}$ subunits can determine the gating dynamics of the GIRK ion channel. As G-proteins are closely associated with GIRK ion channels, we can assume fast binding of the $G_{\beta\gamma}$ subunit to the GIRK ion channel. Under these assumptions, we can equate the normalized G-protein activity with the GIRK ion channel gating parameter $h_{GIRK}$ as summarized in (Eq 9) below.

G-protein activity depends on the activity of the mGluR$_7$ receptor along with other proteins as discussed in Model Conceptualization: Proposed biochemical mechanism and shown in (Fig 2), which self-orgainze to form discrete units of TECs. Since we do not know the number of TECs and their detailed intrinsic dynamics, we choose to model the collective dynamics of TECs and different biochemical interactions within them in an effective way. Hence, instead of using discrete variables for the activity of different biochemicals, we use continuous variables to capture the "average" dynamics of different biochemicals by considering all TECs together.

Our conceptual minimal model aims to reproduce features of the conditional response, namely 1) the conditional response should be independent of CS duration, and 2) changing the dynamics of PKA and G-protein should be sufficient to produce conditional responses of different durations. It considers the four main biochemicals—mGluR$_7$, G-protein, PKA and PP1—and models their overall effective behaviour as observed *in vivo*. Based on the pictorial diagram shown in (Fig 2), the dynamical equations for the proposed biochemical mechanism within individual TECs are as follows:

$$\tau_1 \frac{du}{dt} = \frac{1}{\alpha + x} u(u_0 - u) - \beta u, \tag{6}$$

$$\tau_2 \frac{dv}{dt} = \lambda v(v_1 - v)(v - v_0) - \gamma(w_0 - \delta u)v + I, \tag{7}$$

$$\tau_3 \frac{dx}{dt} = (v - x) \tag{8}$$

$$h_{GIRK} = x/v_0 \tag{9}$$

where $u$, $v$, and $x$ are the activities of PKA, mGluR$_7$ receptor, and G-protein, respectively, while the activity of PP1 is held constant to $w_0$ as per proposed mechanism. In the above model, all the variables along with parameters $\alpha$, $v_1$, $u_0$, $v_0$, $w_0$, and $I$ carry units of $\mu M$. The remaining parameters $\beta$ and $\delta$ are unitless, while $\gamma$ carries units of $\mu M^{-1}$ and $\lambda$ carries units of $\mu M^{-2}$. All parameters and variables are positive including $\tau_i$ for $i = 1, 2, 3$, which have units of milliseconds. Table 1 discusses the biochemical significance of the various terms in (Eqs 6–8). In (Eq 7), the term $-\gamma\delta uv$ denotes the interaction of PKA with mGluR$_7$. Yet, there is no corresponding term in (Eq 6) because such interactions are enzymetic in nature and have very short time scales compared to the response, which we are trying to model. Hence, the activity of PKA does not change when it interacts with other proteins.

**Table 1. Description of the various terms in the minimal model.**

| Term | Description |
|---|---|
| $u(u_0 - u)$ in (Eq 6) | This term models the rise in PKA activity up to its maximum value of $u_0$. It captures the increases in PKA activity due to the rise of [cAMP] by AC activity facilitated by the Calmodulin protein, especially the non-monotonic change in the increase until PKA saturates. |
| $\frac{1}{\alpha + x}$ in (Eq 6) | This term models the net AC activity and captures that upon parallel fiber stimulation, glutamate activates the mGluR$_7$ receptor, which activates G-protein to produce a G$_\alpha$ subunit to block the cAMP molecule production. The specific form is obtained from Hill's equation with Hill's coefficient equal to 1 as only one unit of G$_\alpha$ protein binds to AC. See S2 Text for more details. AC activity activates PKA and that is why we have the product of $u(u_0 - u)$ and $\frac{1}{\alpha + x}$ in (Eq 6). Here $\alpha = K_D$ is the dissociation constant of AC and G-protein binding. |
| $-\beta u$ in (Eq 6) | This term captures the suppressive influence of PDE on PKA activity via hydrolysing cAMP molecules. $\beta$ signifies the strength of the PDE action, which is proportional to its (constant) concentration. Different conditional responses are the result of different PDE concentrations, such that higher PDE concentrations (larger $\beta$) lead to conditional responses of longer duration. |
| $\lambda v(v - v_0)(v_1 - v)$ in (Eq 7) | This term is the effective representation of the 5-state model of the mGluR$_7$ receptor shown in (Fig 2) and captures the switching property of the mGluR$_7$ receptor corresponding to the unaltered conditional response with changing the CS durations. This is achieved by the lowest degree polynomial required to generate an excitable dynamical system behaviour. $v_0$ signifies the (constant) amount of receptors, which are associated with the G-protein. $v_1$ determines the (constant) threshold activity that needs to be crossed to initiate the conditional response, hence $v_0 \gg v_1$. $\lambda$ is set to unity and ensures the correct dimensionality of the term. |
| $\gamma(w_0 - \delta u)v$ in (Eq 7) | This entire term considers the interactions of the mGluR$_7$ receptor with PP1 via $-\gamma w_0 v$ (lowering of receptor activity due to dephosphorylation) and with PKA via $\gamma \delta u v$ (phosphorylation after the conditional response is over). $\gamma$ denotes the (constant) interaction strength of PP1 on the receptor. $\delta$ denotes the (constant) relative strength of PKA and PP1 interactions on the receptor. |
| $I$ in (Eq 7) | This term denotes the rate of activation of the receptor in unit time, which is determined by the strength of the CS in the form of glutamate release from parallel fibers. |
| $v - x$ in (Eq 8) | This term models the G-protein activity as a linear function. This simplification is justified as the G-protein is coupled with the receptor via the RGS8 protein. This means that if the activity of the receptor increases, the G-protein activity increases too. |
| $\tau_1$ in (Eq 6) | Effective time constant for PKA in milliseconds |
| $\tau_2$ in (Eq 7) | Effective time constant for mGluR$_7$ receptor in milliseconds |
| $\tau_3$ in (Eq 8) | Effective time constant for G-protein in milliseconds. Depending on the training, its value can be small or big which results in a short or a long delay in the onset of the conditional response, respectively. |

**Comprehensive mathematical model of proposed biochemical mechanism.** We now present a comprehensive higher-dimensional mathematical model of the biochemical mechanism proposed in Model Conceptualization: Proposed biochemical mechanism based on the full chemical reaction kinetics including detailed biochemical pathways. It allows us to overcome some technical limitations faced by our minimal model. Our minimal model captures the essential interactions between various biomolecules: mGluR$_7$ receptors, PKA, PP1, G-protein using linear and nonlinear terms, which are effective terms but sufficient to qualitatively reproduce the conditional response observed experimentally as we show later. In addition, because of its low dimensionality, it allows us to perform a detailed bifurcation analysis to establish the robustness of the proposed mechanism and our findings. Yet, some of the parameters in our minimal model are effective parameters and, hence, cannot be directly connected to experimentally accessible parameters and molecular interactions. Our full model is able to overcome this limitation. Furthermore, it also allows a more rigorous experimental verification of the proposed biochemical mechanism compared to the minimal mathematical model.

The comprehensive mathematical model comes in the form of mass-kinetic reaction equations not only for $mGluR_7$, PKA, PP1 and G-protein but also their associated proteins and biomolecules considering their detailed biochemical interactions described in Model Conceptualization: Proposed biochemical mechanism. In particular, this model explicitly captures the $mGluR_7$ receptor's 5-state behaviour as depicted in (Fig 2). Depending on the relative concentration of an enzyme compared to its substrate, we have used both Michelis-Menten equations as well as the complete set of enzymetic reactions [65]. Because activities of proteins like PKA will be under regulation by other proteins such as PDE, the total enzyme's (active form) concentration will be a function of time. Also, as the protein interaction with its substrate is considered to be fast, there will be no net decrease in the free protein concentration during its interaction with the substrate. Assuming the total [enzyme] to be $e_0(t)$, the [substrate] to be $s$, the [substrate-enzyme complex] to be $c$ and denoting the [product] as $p$, the mathematical equations for an enzymetic reaction take on the following form

$$\frac{ds}{dt} = -k_1(e_0(t) - c)s + k_2 c \tag{10}$$

$$\frac{dc}{dt} = k_1(e_0(t) - c)s - (k_2 + k_3)c \tag{11}$$

$$\frac{dp}{dt} = k_3 c \tag{12}$$

To simulate such equations, the required $k_1$, $k_2$ and $k_3$ parameter values can be obtained from the experimentally measured values of an enzyme's turnover rate $k_{cat}$ and its affinity for its substrate $k_m$ and from a fixed value of the ratio $\frac{k_2}{k_3} = 4$. This value is recommended in [66] for the ratio as the concentration of the protein complex is low compared to its substrate. Using above standard enzyme reaction kinetic equations in the context of biochemical interactions discussed in Model Conceptualization: Proposed biochemical mechanism, we can derive the comprehensive model describing the collective dynamics of the TEC's. More details including the biochemical reactions and the full set of equations can be found in S3 Text. We only would like to highlight here that based on our proposed biochemical mechanism, the two parameters that control the onset and ISI duration of the conditional response are the concentration of PDE, [PDE], and the rate constant of G-protein activation and deactivation, $k_{gp}$.

**Parameter values.** All parameter values of the comprehensive model are either obtained or constrained based on existing literature and/or experimental observations as provided in S1 Table. However, such constraints cannot be directly translated to all parameters of the minimal model as it uses effective terms to capture multiple biochemical reactions that are individually described in the comprehensive model. Still, many of the parameters in the minimal model have direct counterparts in the comprehensive model and are, hence, experimentally constrained. These parameters are $u_0 = [PKA] = 4.0 \mu M$, $v_0 = [mGluR_7] = 4.0 \mu M$, $w_0 = [PP1] = 4.0 \mu M$, see S2 Table as well as $\alpha = K_D = 0.08 \mu M$ and $\tau_2 = 1/k_{f10} = 10^{-3} sec$ assuming $1 \mu M$ of [Glutamate], see S1 Table Furthermore based on our proposed biochemical mechanism, the parameters $\beta$ and $\tau_3$ in the minimal model can be treated as functions of [PDE] and $1/k_{gp}$ in the comprehensive model, respectively, and they control the models' behavior in similar ways, namely they determine the onset and ISI duration of the conditional response. Indeed, based on all existing studies we are aware of and based on our proposed biochemical mechanism, we have no clear evidence for changes in the value of any other model parameter as a consequence of training with different ISI duration. Therefore as a first approximation, we assume all other

parameters to be independent of ISI duration. Regarding the remaining effective parameters of the minimal model, we chose their specific values to match the conditional response dynamics of the comprehensive model. Specifically, the values of $\tau_1$ and $\tau_3$ are chosen to match the time scales of the minimal model's dynamics to those of the comprehensive model. Similarly, $v_1$ controls the threshold value for $I$ to initiate the conditional response. Since the conditional response behaviour is independent of stimulus duration and corresponds to that of an excitable system, we can use small and constant values for both $v_1 = 0.05\mu M$ and $I = 0.1\mu M$. For the stimulus duration, a short duration of about 20msec is used as applied experimentally in [13] while an arbitrary long stimulus duration is used to show that the conditional response is indeed independent of stimulus duration. The choice and range of the remaining parameters, namely $\beta$, $\gamma$, $\delta$ and $\lambda$, will be discussed in Robustness and stability of minimal model.

**Numerical simulations.**   We use the odeint module of python's scipy library to integrate the ordinary differential equations of our mathematical models to establish their behavior. The odeint module uses lsoda from the FORTRAN library odepack and solves the initial value problem for stiff or non-stiff systems of first order ordinary differential equations. In particular, lsoda automatically selects the appropriate integration method and step size for a given (stiff or non-stiff) system of ordinary differential equations [67]. The behavior of the minimal model does not depend on the specific choice of the initial conditions as long as they are physically relevant, i.e., $0 < u, v, x < 4$ since there is only one stable fixed point, see Robustness and stability of minimal model for more details. For the comprehensive model, we use the initial conditions given in S2 Table.

## Results

We now discuss and compare the dynamics of various biomolecules essential for the conditional response using both the minimal and the comprehensive model. We analyze the robustness and parameter sensitivities for both our mathematical models and subsequently we establish a number of predictions that can be tested experimentally.

### Properties of both minimal and comprehensive mathematic models

Since experimental results have shown that the conditional response is independent of CS durations, the activation of the G-protein must also satisfy this property as it regulates GIRK ion channels [22]. This behavior is indeed captured by our mathematical models. As shown in (Fig 3), they also successfully capture the dynamics of the biochemicals—PKA, mGluR$_7$, and G-protein—of the proposed biochemical mechanism.

As per our proposed mechanism, before CS, PKA activity is high while the activity of the mGluR$_7$ receptors and G-proteins are low. Upon CS, mGluR$_7$ receptors become active, which in turn activate G-proteins. Due to activation of G-proteins, PKA activity drops, which causes the deactivation of mGluR$_7$ receptors by PP1. This leads to the deactivation of the G-proteins by RGS8. When the stimulus is turned off, the activity of the various proteins returns to the original state as shown in the top panel of (Fig 3). However, if the stimulus remains on for a long time, then even very low G-protein activity can prevent the rise of PKA activity to a value that would be high enough to overcome the activity of PP1. As a result, PP1 activity will be crucial to cause dephosphorylation of mGluR$_7$ receptors and to block their active sites to prevent the initiation of another conditional response as shown in the middle panel of (Fig 3). Very faint G-protein activity in case of long duration CS can be observed in the bottom panel of (Fig 3). This is enough to prevent reactivation of the conditional response.

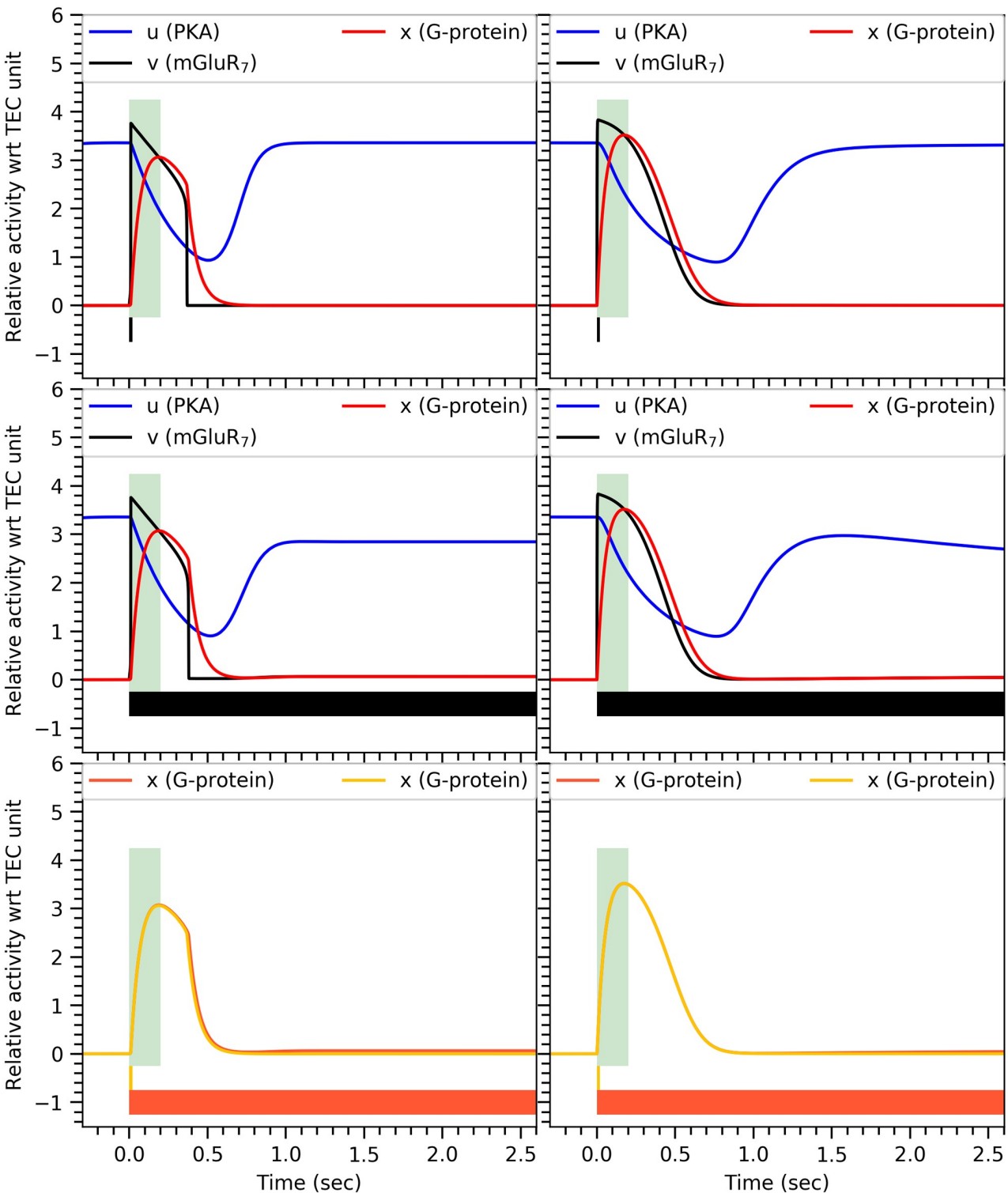

**Fig 3. Temporal behaviour of PKA, mGluR$_7$, and G-protein.** Time varying quantities of PKA, mGluR$_7$ and G-protein obtained from the minimal model (left) and the comprehensive model (right) upon short 20msec (top) and long >2.5sec (middle) stimulus durations represented by the horizontal bar at the bottom of each panel. The green vertical bar represents ISI duration (200msec in all cases). In the bottom panels, activity of the G-protein is shown as in the upper panels but for the two different stimulus durations together, indicated by the two color-coded horizontal bars at the bottom. Both responses are almost identical implying that the G-protein activity is indeed independent of stimulus duration. Parameters values for our minimal dynamical model are: $\beta = 8.5$, $\delta = 1.0$, $\gamma = 1.4\mu M^{-1}$, $\lambda = 1.0\mu M^{-2}$, $\tau_1 = 2500msec$, $\tau_3 = 58msec$, while the remaining parameter values are given in Parameter values. For the comprehensive model, [PDE] = $1.25\mu M$ and $1/k_{gp}$ = 50msec, while the remaining parameter values are given in S1 Table.

## Robustness and stability of minimal model

The robustness of the proposed mechanism and the dynamics of the minimal mathematical model shown in (Fig 3) can be established by a phase space and bifurcation analysis. We first discuss the relevant fixed point of the model based on the nullclines before presenting a comprehensive bifurcation diagram.

**Nullclines.** Our model has three different types of nullclines, one for each variable. For $x$, the nullcline ($\partial x/\partial t = 0$) has only a single trivial solution corresponding to $x = v$, see (Eq 8). Thus, we can focus on $u$ and $v$ in the following. For $I = 0$, as (Fig 4) shows, there are two $u$-nullclines (where we have used the $x$-nullcline $x = v$ in (Eq 6) and two $v$-nullclines, including $u = 0$ and $v = 0$. The latter two ensure that ($u$, $v$, $x$) never become negative *aka* unphysical. The highlighted intersection points between nullclines in (Fig 4) correspond to the fixed points of our model: There is only one physically relevant stable fixed point with $v = 0$ and $u > 0$, which describes the "resting" state of the Purkinje cell after training/learning is completed as discussed in the section Model Conceptualization: Proposed biochemical mechanism. For $I \neq 0$ in (Fig 4), the $v$-nullclines change and the location of the stable fixed point moves to a lower value of $u$. The dynamics takes the system from the previous location of the fixed point to the new location along a long trajectory. The duration corresponds to the learned time interval and is encoded in the values of $\beta$ and $\tau_3$. The parameter $\beta$ controls the slope of the linear $u$-nullcline and, hence, the specific location of the stable fixed point as well as the rate at which PKA activity is reduced. The $\tau_3$ parameter controls the rate of change in G-protein activity and, hence, the opening and closing of GIRK ion channels associated with the decrease in firing rate of the Purkinje cell. Consequently, it determines the time delay between the onset of the conditional response and the minimum firing rate. We would like to note that there is only a finite range of conditional responses that can be obtained from the minimal model for a fixed set of parameter value. In the presence of a continuous stimulus (non-zero value of $I$) as $\beta$ decreases, the $u$ value of the stable fixed point rises until it reaches the apex of the $v$-nullcline. At this point, a stable limit cycle emerges. As this behaviour is not observed experimentally, this sets the lower limit for suitable values of $\beta$. For the upper limit, an increase in $\beta$ lowers the $u$ value of the stable fixed point and, hence, requires a stronger minimal stimulus to initiate the conditional response such that the stimulus strength $I$ determines the maximum suitable value of $\beta$.

The remaining two free parameters are $\gamma$ and $\delta$. They affect the dynamics of the model, for example, by altering shapes and positions of $u$- and $v$-nullclines similar to $\beta$. For instance, $\gamma$ and $\delta$ widen the quadratic $v$-nullcline and bring it closer to the $u = 0$ nullcline. This will affect the existence, location and stability of the fixed points including the relevant one corresponding to the "resting" state. For example, its stability changes whether it lies inside (unstable/saddle) or outside (stable) the quadratic $v$-nullcline as the signs of the flow in (Fig 4) show. The mechanism we propose for the time-encoding memory formation in the Purkinje is directly tied to the qualitative layout of the nullclines in (Fig 4) and the resulting stability of the relevant fixed point. Thus, the values of the parameters $\beta$, $\gamma$ and $\delta$ are naturally constrained and directly tied to the biological interpretations associated with the model's dynamics. In particular, to ensure the stability of the relevant fixed point, we need to choose the values of $\gamma$ and $\delta$ within certain ranges to position the $u$- and $v$-nullclines appropriately. As the value of $\beta$ determines the specific ISI durations and, hence, needs to be varied, we also need to make sure these variations in $\beta$ do not change the properties of the relevant fixed point either. Besides controling the shape and position of the quadratic $v$-nullcline, $\gamma$ also influences the rate of decay of mGluR$_7$ receptor's activity. For numerical values of $\gamma$

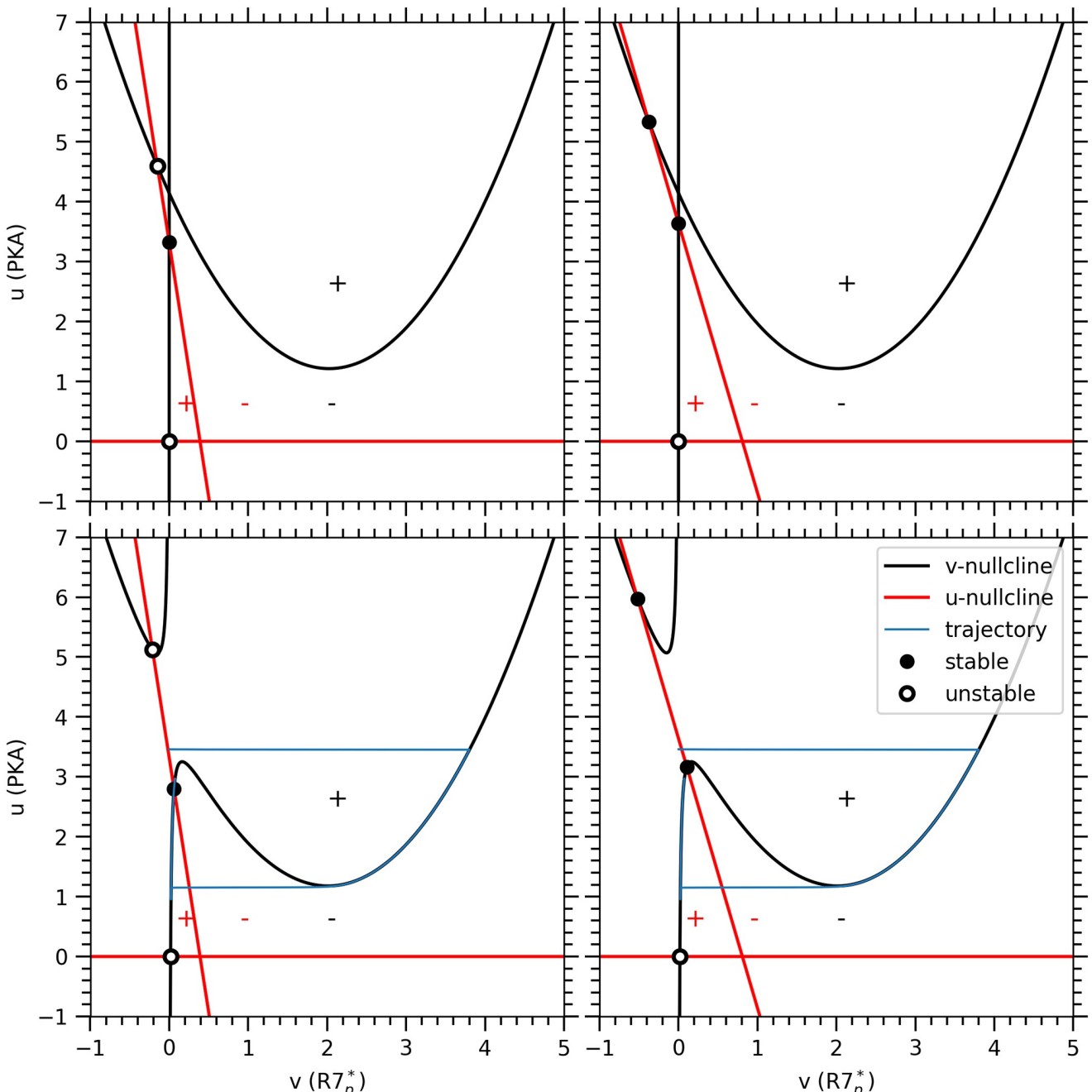

**Fig 4. U and V nullclines for different parameters.** Using the $x$-nullcline $x = v$ in (Eq 8), we obtain a pair of nullclines for both variables $u$ and $v$ shown for the same parameters as in (Fig 3) except for $\beta$ and $\tau_3$, which are chosen to correspond to different ISIs (see Table 2): Top panels from left to right are for ISI = 200msec, 400msec with $I = 0\mu M$. Bottom panels from left to right are the same but with $I = 0.1\mu M$. These panels also include the projection of the trajectories from the location of the stable fixed point without stimulus ($I = 0\mu M$) to the new one ($I = 0.1\mu M$). The time it takes to get from one to the other corresponds to the ISI duration. Closed circles represent stable fixed points, while open circles represent unstable/saddle fixed points. +/− signs represent the signs of $du/dt$ and $dv/dt$, which change across $u$ and $v$-nullclines, respectively.

close to

$$\gamma > \frac{(v_0 - v_1)^2}{4w_0} = 0.975 \mu M^{-1}, \tag{13}$$

the minimum of the quadratic $v$-nullcline approaches the $u = 0$ nullcline. Physically, this means that as the interaction strength between the receptor and the two proteins, PKA and PP1, decreases for smaller $\gamma$, the state of the receptor can only be changed if the difference between PKA and PP1 activity is large. Thus, in the presence of constant PP1 (fixed $w_0$), the PKA activity, $u$, has to take on lower values. As a decrease in PKA activity is regulated by the action of PDE on PKA, a decrease in PKA also slows down the *rate* of change in PKA activity. Slower changes in $u$ also slow down the variation in $v$ as the receptor activity will not change until PKA reaches its minimum value, which corresponds the minimum of the quadratic $v$-nullcline. Overall, this extends the duration during which the mGluR$_7$ receptor remains active and, hence, the duration of the conditional response. On the other hand, larger numerical values of $\gamma \gg 1 \mu M^{-1}$ raise the minimum of the quadratic $v$-nullcline, which implies that even a small difference between PKA and PP1 activity can change the state of the receptor. In such a case, PDE can quickly down regulate PKA activity, which leads to faster deactivation of the receptor. More importantly, if the inequality in (Eq 13) is violated, a new pair fixed points emerges, one stable and one unstable. This must be avoided in order to be consistent with our proposed mechanism. As a result, we chose an intermediate value of $\gamma = 1.4 \mu M^{-1}$ in (Fig 4), which produces a sufficiently wide window of conditional responses as observed in the experiments. Linear stability analysis shows that for

$$\delta < \frac{v_1 v_0 + \gamma w_0}{\gamma u_0} \tag{14}$$

the relevant fixed point is *always* stable. This is why we chose $\delta = 1.0$ in (Fig 3), which satisfies (Eq 14) for any arbitrarily large value of $\gamma$. However, if $\delta$ is significantly decreased, then a higher value of $I$ is required to initiate the conditional response. From a biological perspective, PP1 dominates over PKA relatively for smaller values of $\delta$ and hence this requires a stronger stimulus to initiate the conditional response. In summary, the possible range for $\delta$ is $0 < \delta \leqslant 1$ while for $\gamma$ it is $0.975 \mu M^{-1} < \gamma$.

**Bifurcation analysis.** To obtain the full bifurcation diagram, we use the numerical bifurcation software MATCONT [68], which provides us with both local and global bifurcations for our model. Specifically, we find codimension-1 (codim1) bifurcations including branch point (transcritical), limit point (saddle-node) and Hopf bifurcations, rare codimension-2 (codim2) bifurcations including Bogdanov-Takens, Generalised-Hopf and Cusp bifurcations as well as global bifurcations such as homoclinic bifurcations. (Fig 5) shows the fixed points of our model and the various local codim1 and global bifurcations over certain parameter ranges. For $\delta = 1.0$ and both values of $\gamma$, the fixed point with $v = 0$ and $u > 0$, which describes the "resting" state of the Purkinje cell after training/learning is completed, is stable for all values of $\beta$ as indicated by the straight solid line. In contrast, for $\delta = 1.2$ this is only true for large values of $\beta$. When decreasing $\beta$, it becomes a saddle point through a branch point. Over a very limited range in $\beta$ the new stable fixed point first loses its stability through a supercritical Hopf bifurcation giving rise to a stable limit cycle, which then is destroyed through a homoclinic bifurcation around $\beta = 6.7645$, turning the original relevant fixed point into the attractor of the physically relevant dynamics despite being a saddle point. One way to verify this is that there exists no stable fixed point for $\delta = 1.2$, $1.5 \lesssim \beta \lesssim 6$, yet the directions of all flow vectors at the surface of a cuboid of sides $u = 4, v = x = 5$ starting from the origin, point inwards. This implies

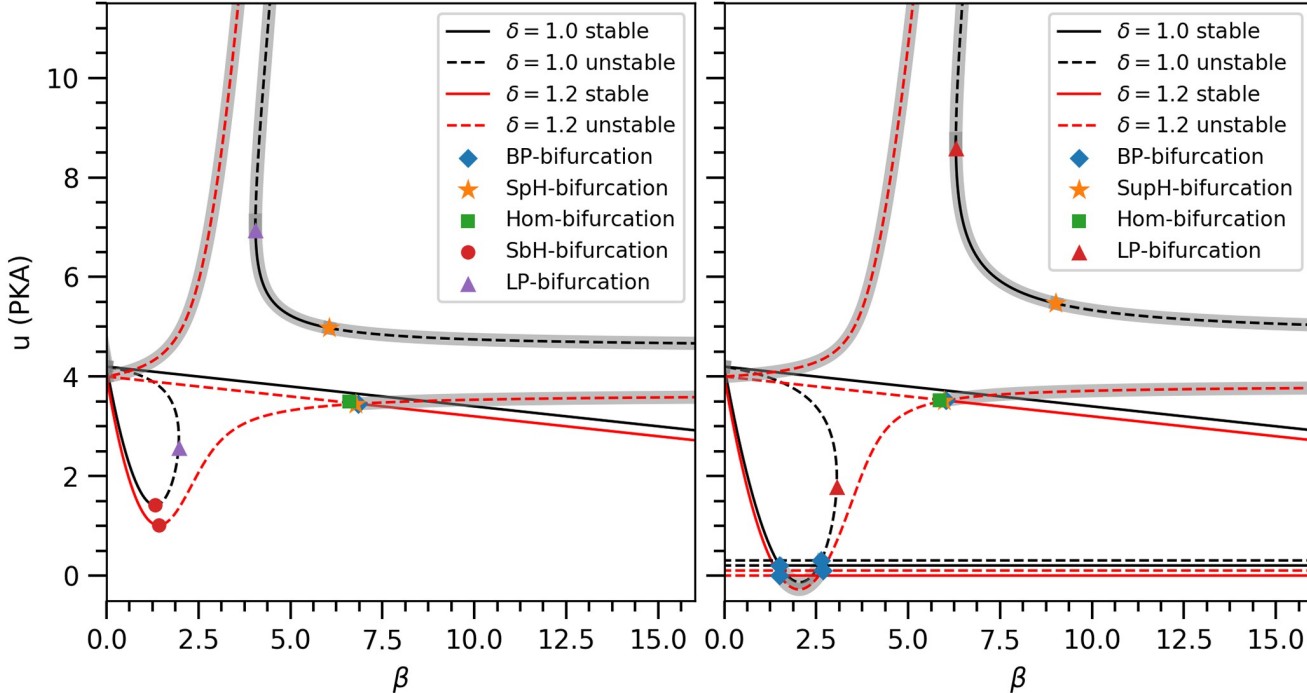

**Fig 5. Stability of fixed points and their bifurcations for different values of $\beta$, $\gamma$ and $\delta$.** Abbreviations for various bifurcations as BP: Branch point, SpH: Supercritical Hopf, SbH: Subcritical Hopf, Hom: Homoclinic, LP: Limit point. Each curve corresponds to one fixed point solution for given set of parameters values. Solid lines represent stable fixed points, while dashed lines represent unstable or saddle fixed points. The black curves in both panels are shifted upwards to differenciate them from the red curves as they would overlap otherwise. We ignored the fixed point (0,0,0) as it remains unstable for all $\gamma > 0$, $\delta > 0$ and $\beta < u_0/\alpha >> 1$. Unphysical solutions are shaded with a gray background. **Left panel** ($\gamma = 1.4\mu M^{-1}$): For $\delta = 1.0$, the relevant fixed point is stable for all shown values of $\beta$ values. For $\delta = 1.2$, the same fixed point is only stable for larger values of $\beta$. When decreasing $\beta$, it becomes a saddle point through a branch point. Over a very limited range in $\beta$ the new stable fixed point first loses its stability through a supercritical Hopf bifurcation giving rise to a stable limit cycle, which then is destroyed through a homoclinic bifurcation around $\beta = 6.7645$, turning the original relevant fixed point into the attractor of the physically relevant dynamics despite being a saddle point. This continues to be the case until a new stable fixed point emerges through a sub-critical Hopf bifurcation at much lower values of $\beta$. **Right panel** ($\gamma = 0.9\mu M^{-1}$): In addition to the fixed points in the left panel, a new pair of fixed points with $u = 0$ is present for this smaller value of $\gamma$ since the quadratic $v$-nullcline is shifted down. For clarity, we have shown them as separate lines (two for each value of $\delta$). One of them is stable over a large range of $\beta$. The pair of new fixed points undergoes their own bifurcations at $\beta = 1.5076$ and $\beta = 2.6795$, respectively, replacing the sub-critical Hopf bifurcation.

that an attractor exist within the cuboid. The aforementioned limit cycle is the attractor over the very limited parameter range of its existence, leaving the original relevant fixed point as the attractor otherwise. This continues to be the case until a new stable fixed point emerges through a sub-critical Hopf bifurcation at much lower values of $\beta$.

While this suggests that larger values of $\delta$ violating (Eq 14) would also be largely consistent with our proposed mechanism, this is not the case. Specifically, in the absence of PDE, i.e., for $\beta = 0$, the only stable fixed point would be $v > 0$ and $u > 0$—even in the *absence* of a stimulus. This implies that receptors can become active and initiate the conditional response on their own without a stimulus being provided, which contradicts the experimental observation [12]. If $\delta$ does not violate (Eq 14), there are two stable fixed points for $\beta = 0$: $u > 0$ and $v = 0$; $u > 0$ and $v > 0$. Being an excitable model, any initial condition within the quadratic $v$-nullcline will settle to the active state, i.e., $u > 0$ and $v > 0$, while if outside the $v$-nullcline, the final state would be the inactive state, i.e., $u > 0$ and $v = 0$, corresponding to the relevant fixed point of our model. As the shape of the quadratic $v$-nullcline depends on the values of $\gamma$ and $\delta$, the region of this bistability in $\beta$ becomes arbitrarily small for sufficient large values of $\gamma$ such that only the relevant stable fixed point remains.

As mentioned above, if the inequality in (Eq 13) is violated, an additional pair of fixed points emerges. This is the case for $\gamma = 0.9$ (right panel in (Fig 5), where we have an additional stable fixed point with $u = 0$ for large values of $\beta$. From the biological perspective, this corresponds to a weak interaction between the receptor and PKA and PP1 giving rise to partially active receptors in the absence of PKA activity. Due to weak interactions, PP1 is not effective in deactivating the receptor's activity, which means that once the receptor gets into this state, it will remain in the state. Even if we change the value of $\beta$, it will not go back to its inactive state. A condition like this where PKA or PP1 interact weakly with their target protein is not biochemically realistic as it offers no advantage to have them close to each other as observed and argued in previous studies [31–33]. This fixed point becomes unstable at $\beta \sim 1$ through a branch point and a new stable fixed point emerges corresponding to the active state discussed above with finite residual activity for both PKA and the receptor. In summary, the relevant fixed point with $v = 0$ and $u > 0$, which describes the "resting" state of the Purkinje cell after training/learning is completed, is the unique stable fixed point of our model as long as $0.975 \mu M^{-1} < \gamma$, $0 < \delta \leqslant 1.0$ and $0.0 \lesssim \beta < u_0/\alpha = 50.0$. Note that at $\beta = u_0/\alpha$, $u$ becomes zero. This shows that the desired model behavior is quite robust against changes in $\gamma$ and $\beta$ and to a somewhat lesser degree in $\delta$ parameter. In particular, the following properties of the model remain preserved: i) The G-protein activation remains largely independent of the CS duration, and ii) no oscillatory response emerges. These two features are essential in order to reproduce the observed experimental results.

## Sensitivity analysis of the comprehensive model

In the previous section, our bifurcation analysis revealed not only the robustness of the dynamics of the minimal model but it also highlighted that all its properties are in line with the proposed biological mechanism. While we cannot do such an analysis of the comprehensive model due to its high-dimensionality, a simple local parameter sensitivity analysis can establish its robustness to express the conditional response. Specifically, our sensitivity analysis aims to find the individual range of each parameter over which the opening of GIRK ion channels for the conditional response of the Purkinje cell occurs. As GIRK ion channels can fully open when 3-4 G-protein subunits bind to them [69], we select the G-protein activity averaged over all TECs as our indicator and require it to be between 3 and 4. This analysis shows that most of the kinetic parameters of the comprehensive model are fairly robust as their values can be increased or decreased by a factor of 50. However, there are a total of four parameters that can only be varied less than 5-fold with respect to their original values. These more sensitive parameters are: $k_{gp}$, $k_{f11}$, $k_{cat7}$ and $k_{cat8}$, see S3 Table. As mentioned before, $k_{gp}$ controls the rate of G-protein activation and deactivation and, hence, both onset and delay of the conditional response. $k_{f11}$ controls the rate of dephosphorylation of the receptor by PP1. Based on the experimental evidence discussed in Model Conceptualization: Proposed biochemical mechanism, $k_{f11}$ is considered to be constant for all different conditional responses and the range over which it allows for the conditional response to be observed is consistent with the expected range of rate constants, see S3 Table for more details. $k_{cat7}$ and $k_{cat8}$ control the cAMP production and hydrolysation by AC and PDE enzymes, respectively. The sensitivity with respect to $k_{cat7}$ and $k_{cat8}$ is not concerning since they are constrained by their experimentally measured values and we assume them to be constant in the absence of literature indicating otherwise.

Besides the kinetic parameters, the concentration of biomolecules can also vary in principle. Yet, the only biomolecule whose concentration could vary in our case is PDE as the reactions stoichiometry constrains the concentrations of all other biomolecules. Indeed, the comprehensive model's dynamics depends sensitively on [PDE], see S3 Table. This is not surprising since

based on our proposed biochemical mechanism [PDE] controls both onset and duration of the conditional response.

## Neuronal spiking activity of the Purkinje cell

Now we focus on the conditional response dynamics of the Purkinje cell, where our mathematical model of the proposed biochemical mechanism—either the minimal one or the comprehensive one—determines the dynamics of the gating parameter $h_{GIRK}$, see Purkinje cell model. In (Fig 6) we show that the suppression of firing rates during the conditional response of ISI = 200msec is independent of CS durations—for both the minimal model and

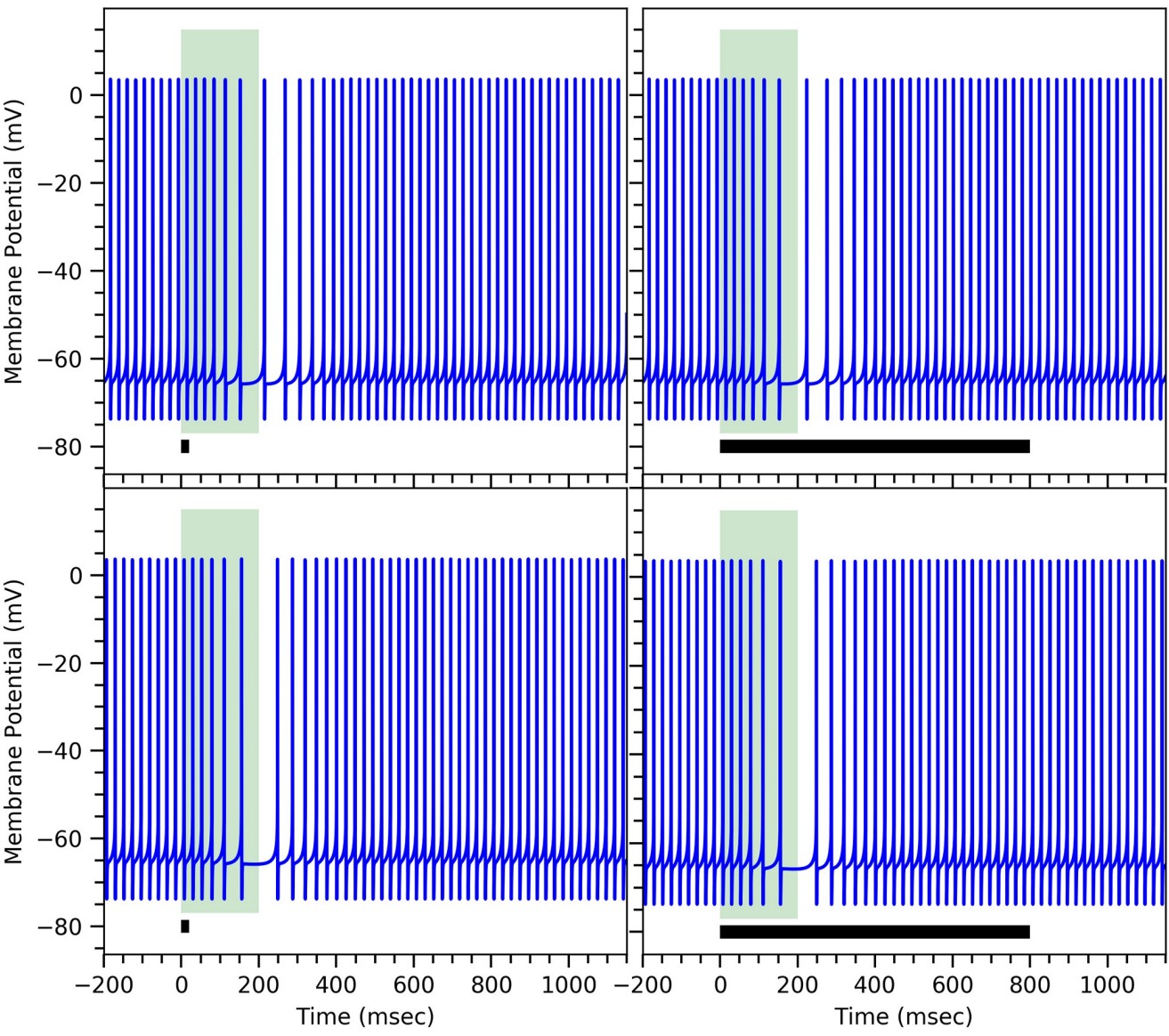

**Fig 6. Conditional response in both models is independent of conditional stimulus duration.** Membrane potential of the Purkinje cell during conditional response for ISI = 200msec obtained using our minimal model (top panels) and our comprehensive model (bottom panels). The width of the light green vertical bar corresponds to the duration of the ISI and the black bar at the bottom signifies the conditional stimulus duration, which is short in the left panels and long in the right panels. Parameters for both models are the same as in (Fig 3).

**Table 2. Model parameters for different conditional responses of the Purkinje cell.**

| ISI (msec) | Minimal Model | | Comprehensive model | |
|---|---|---|---|---|
| | $\beta$ | $\tau_3$ (msec) | [PDE] in $\mu$M | $1/k_{gp}$ in (msec) |
| 200.0 | 8.5 | 58.0 | 1.25 | 50.0 |
| 300.0 | 6.1 | 97.0 | 0.98 | 78.7 |
| 400.0 | 4.7 | 139.0 | 0.88 | 100.0 |

comprehensive model—as observed in the experiments [12]. Experimentally, Purkinje cells have shown a drop of roughly between 10% and 30% in their firing rate [13], while both our models show a drop in the firing rate between 20% and 30% for the chosen value of g$_{GIRK}$. Note that the dynamics shown in (Fig 6) can be considered as an average response of the firing rate during the conditional response given the deterministic nature of our model.

In order to obtain a conditional response of longer duration, more mGluR$_7$ receptors need to be inserted into the synapse. These extra receptors cause a rise in the value of $\tau_3$ (minimal model) or $1/k_{gp}$ (comprehensive model) and lower the value of $\beta$ *aka* the PDE concentration as discussed earlier in Model Conceptualization: Proposed biochemical mechanism. Different values of $\tau_3$, $1/k_{gp}$, [PDE] and $\beta$ corresponding to different conditional responses are summarized in Table 2. (Fig 7) shows different long duration conditional responses, which match with the experimental results [12]. For ISI = 200msec as shown in (Fig 7)(left panels), the firing rate drops and then rises slowly, which is consistent with the experimental results. For higher ISI = 400msec, the drop and rise of the firing rate is observed to be even slower compared to ISI = 200msec as shown in (Fig 7)(right panels). Again both our minimal and comprehensive models produce similar results. Both of our models, in fact, can produce conditional responses of ISI = 100msec and ISI = 1000msec, which are the extreme values observed experimentally [42, 70]. This is simply because $\beta$—or equivalently [PDE]—and $\tau_3$—or equivalently $1/k_{gp}$—can take on a wide range of values without fundamentally changing the behavior. This can be seen, for example, in the bifurcation plot (Fig 5) for $\delta = 1.0$ and $\gamma = 1.4$. Larger values of $\beta$ imply shorter ISI durations of the conditional response and vice versa.

To summarize, changing values of both $\beta$ and $\tau_3$ (minimal model) or $k_{gp}$ and [PDE] (comprehensive model) simultaneously allows us to model different conditional responses within our mathematical framework. We would like to point out that changing either one of the two alone does not reproduce the experimental behavior.

The three normalized conditional response firing patterns obtained from both our models shown in the left panels of (Fig 8) match with the experimental results [13]. Moreover, we also determine the delays in the onset of the conditional response by recording the time when the normalized firing rate drops to 95% of the spontaneous firing rate. These recorded onset delays are consistent with the experimental observed values [12, 13], see Table 3. Such variable onset delays can be observed in certain species such as rabbits and ferrets [12, 71] but there are other species such as mice that exhibit a fixed onset delay of conditional eye-blink response [72]. Our models can reproduce such a behavior as well if the individual G-protein subunits are strongly interacting with each other—as sometimes observed experimentally [73]—such that in (Eq 2) the 4 units of G-protein subunits, $G_{\beta\gamma}$, give rise to a term $h_{girk}$ instead of $h_{girk}^4$. This is shown in the bottom panels of (Fig 8) using the very slightly modified parameter values given in S4 Table

Our proposed mechanism also explains why the time-memory remains unaffected in the presence of mGluR$_7$ antagonist MMPIP as observed in the experiment [13]. Specifically, because of the presence of MMPIP, fewer mGluR$_7$ receptors are left to activate GIRK ion

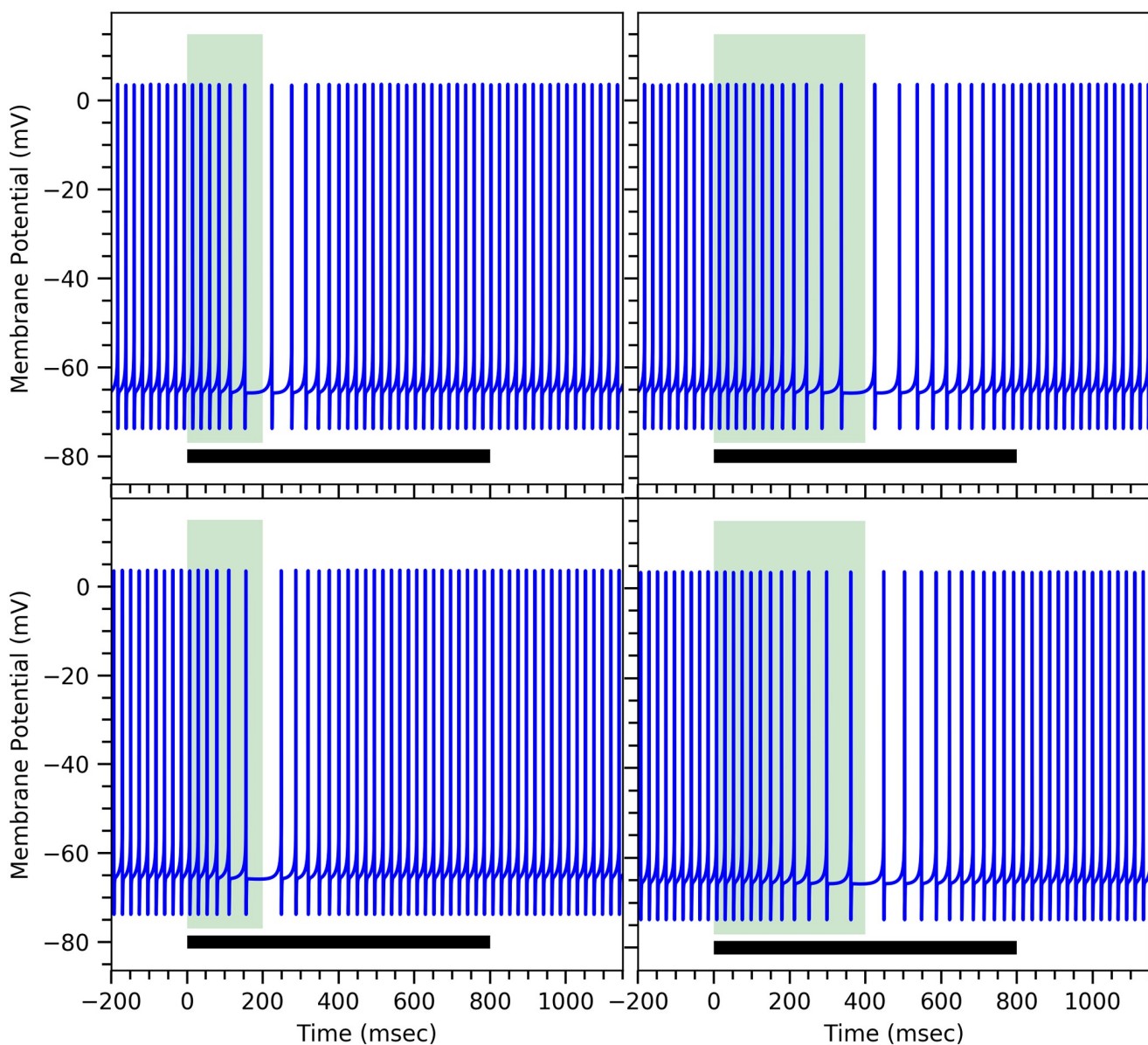

**Fig 7. Different conditional responses of the Purkinje cell obtained from the mathematical models.** Membrane potential of the Purkinje cell during conditional response for different ISIs = 200msec (left panels) and 400msec (right panels) for the minimal model (top panels) and the comprehensive model (bottom panels). The specific parameters values are given in Table 2, all others are the same as in (Fig 3). The width of the light green vertical bar corresponds to the duration of the ISI interval. The black horizontal bar at the bottom represents the conditional stimulus duration.

channels, which leads to a smaller drop in firing rate. However, reducing the net amount of active mGluR$_7$ does not inhibit the internal interactions between the receptor and other proteins involved in our proposed mechanism. Hence, the time-memory, which is encoded within effective dynamics of the biochemical reactions, is unaffected by MMPIP as shown in (Fig 8) (right panels in top and middle rows). Note that in (Fig 8) the action of an increasing dose of MMPIP is simulated by decreasing the value of the parameter $g_{GIRK}$. As the corresponding values of $g_{GIRK}$ have not been measured experimentally as mentioned in Purkinje cell model, we choose suitable values of $g_{GIRK}$.

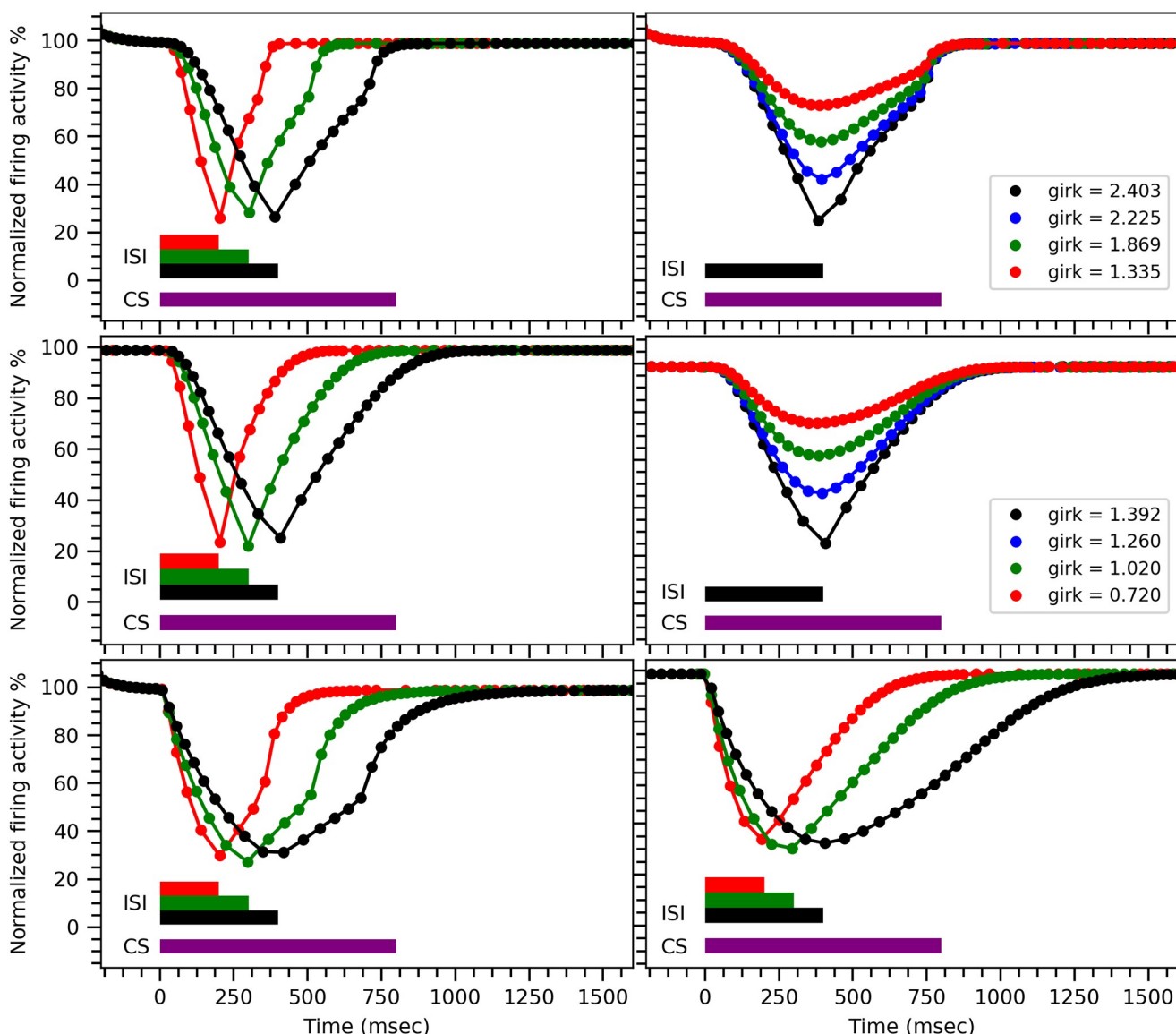

**Fig 8. Conditional response profiles for different ISIs and different amounts of MMPIP.** Conditional response profiles obtained from the minimal model (top panels) and the comprehensive model (middle panels) for different parameter values (see Table 2, all other parameters as in (Fig 3)) (left panels), and in the presence of the mGluR7 receptor's antagonist MMPIP (right panels). As a mGluR7 antagonist, MMPIP leads to a decrease in the net amount of active mGluR7 and, hence, the amount of active GIRK ion channels, which corresponds to smaller values of $g_{GIRK}$ (see Eq 2). The bottom panels show the conditional response profile for a modified G-protein subunit dynamics (see text for details) in the minimal model (left panel) and in the comprehensive model (right panel). Note that the normalized instantaneous firing activity is calculated here by taking the inverse of the time interval between two successive spikes, centered on the midpoint between the two spikes, and dividing it by the firing frequency before the onset of the conditional response.

**Table 3. Onset delay for different conditional responses of the Purkinje cell.**

| ISI (msec) | Onset delay (msec) | | Experimental data (msec) [12, 13] |
|---|---|---|---|
| | **Minimal** | **Comprehensive** | |
| 200.0 | 52.0 | 40.0 | 48.0±34 |
| 300.0 | 71.0 | 60.0 | 73.0±18 |
| 400.0 | 95.0 | 80.0 | 90.0±20 |

## Model predictions

Based on our proposed models, we can make two predictions that can easily be tested by experiments. 1) If PP1 is knocked out then active mGluR$_7$ receptors will never deactivate once they have been activated by CS and, hence, the G-protein will remain active. This implies that the Purkinje cell will not fire again after receiving CS as shown in (Fig 9). 2) On the other hand, knocking out PKA activation will allow PP1 to dephosphorylate mGluR$_7$ receptors and, hence, the G-protein cannot be activated. This implies that the Purkinje cell will not exhibit a conditional response as shown in (Fig 9). As both our mathematical models produce the same predictions, we only show the results for the minimal model in (Fig 9).

However, in reality biological cells are very robust and have redundancy mechanisms to overcome such behaviours. As a result, there might be still a weak conditional response

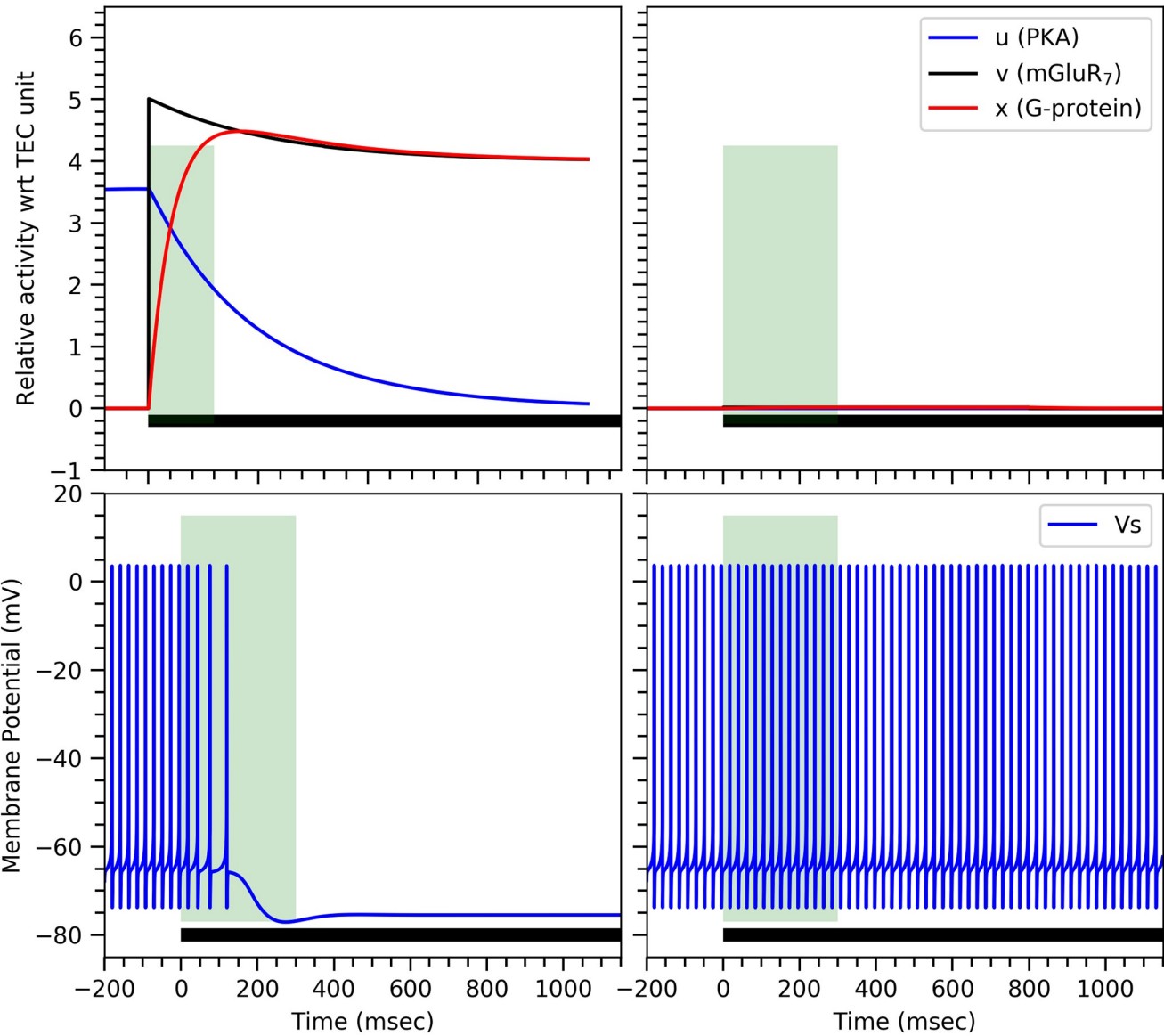

**Fig 9. Model predictions for knockout experiments.** In our minimal model, PP1 can be knocked out by setting $w_0 = 0.0$ at the onset of stimulus (top left panel), which prevents the Purkinje cell to fire again after the initiation of the conditional response (bottom left panel). PKA can be knocked out by setting $u_0 = 0.0$ in our model (top right panel), which prevents the Purkinje cell to initiate a conditional response (bottom right panel).

observed after knocking out PKA or a slow deactivation of G-protein after knocking out PP1, but in both cases significant effects on the conditional response should be observed.

## Specific experimental options to test the proposed model

There are various experimental options to check whether our proposed mechanism for the conditional response is valid or not, including the two model predictions mentioned above.

1. As PP1 desensitizes the $mGluR_7$ receptor during conditional response, blocking of PP1, using Okadaic acid, for example, must affect the deactivation rate of GIRK ion channels during the conditional response. This would test the first model prediction. Alternatively, one could block the PDE enzyme via IBMX (3-isobutyl-1-methylxanthine) since without a drop in PKA activity, PP1 cannot dephosphorylate the receptor.

2. As PKA is an essential biochemical for the resensitization of the receptor and maintaining low PP1 activity, reducing PKA activity in the cell will prevent the Purkinje cell from suppressing its firing rate as PP1 will desensitize the receptor and therefore GIRK ion channels will not be activated. This can be verified by using, for example, cAMPS-Rp or triethylammonium salt, which will block the cAMP production and, hence, PKA. This would test the second model prediction. Alternatively, one could block the AC enzyme via SQ22536 since without activity of AC, PKA cannot be activated.

3. If $mGluR_1$ receptors are activating PKC then blocking of $mGluR_1$ receptors using CPCCOEt during training will not initiate trafficking of $mGluR_7$ receptors and thus no conditional response should be observed even after extensive training.

4. Use of RGS8 knockout specimen should allow only long duration conditional response: Without RGS8 protein, the activation and deactivation of G-protein will be much slower and will produce only long conditional response durations.

## Discussion and conclusion

As both the minimal and the comprehensive mathematical models agree very well with the experimental results, we conclude that our proposed biochemical mechanism can successfully reproduce the conditional response features: 1) Temporal profile of firing rate for different ISI durations, 2) its independence of CS duration, and 3) the various behaviors of the onset delay of the conditional response with changing ISI duration. In addition to these, our models are also able to capture the effect of blocking $mGluR_7$ receptor on the conditional response consistent with experimental observations. In particular, our proposed mechanism makes consistent statements regarding how ISI duration should affect the training period duration. Furthermore, our mathematical models and proposed biochemical mechanism are applicable to both trace and delayed conditioning. This is because at the level of individual cells, both types of conditioning engage the same Purkinje cells, which show similar conditional behaviour in the presence of CS [74]. Yet, some model limitations do remain, which should to be addressed in the future after experimental results further validate our proposed mechanism. These include:

1. Both mathematical models assume simple activation and deactivation dynamics for the $G_\alpha$ and $G_{\beta\gamma}$ subunits depending on receptor's activity even though individual G-protein subunits have been shown to modulate the GIRK ion channel gating dynamics [73]. Due to the slower dynamics of the modulation compared to the conditional response, we have neglected this here. However, RGS8 can potentially accelerate the dynamics of the G-

protein subunits such that a more complex conditional response behaviour could arise including the double pause conditional response observed in experiments [75].

2. While both mathematical models can fully capture the conditional response *after* training is completed, they do not directly describe the learning process itself. A mathematical model for the learning process based on our proposed biochemical mechanism would need to capture aspects of learning experimentally observed during training such as the increase of the ISI duration of the conditional response until it reaches its final value [17]. Such a behavior would require the adjustment of $k_{gp}$ and [PDE] in the comprehensive model or $\tau_3$ and $\beta$ in the minimal model over time. Modeling this aspect from a biological perspective is highly nontrivial as it involves the translocation and protein complex formation of the various proteins involved in the proposed mechanism. Hence, it remains a challenge for the future.

3. By construction, both mathematical models also do not capture the dynamics of individual TEC units. Instead, they model the collective behaviour of all TEC units. This simplification, however, fails to explain how different decoupled TEC units can produce a strong and robust conditional response of a specific duration. cAMP biomolecules can potentially offer a necessary coupling mechanism as merging of cAMP microdomains [76] from different TEC units can synchronize the dynamics across different TECs and collectively produce a conditional response of a specific duration.

In terms of the bigger picture, we introduced a potential biochemical mechanism to explain time-encoding memory formation within a *single* synapse of a Purkinje cell. This time-encoding memory is stored in an excitatory synapse, but it is associated with an inhibitory response, i.e., the suppression of the Purkinje cell's tonic firing rate in the presence of an excitatory stimulus, namely glutamate discharge from the parallel fiber. During conditional training, Purkinje cells imprint the time information by expressing an appropriate amount of mGluR$_7$ receptors on the synapse, while encoding time information in the form of effective dynamics of biochemical interactions. The memory is stored by forming a protein complex we call TEC. Alterations of effective dynamics within TECs will change its temporal signature, while the removal of receptors from the synapse will cause memory loss. However, during retraining, the previous memory can quickly be reacquired and it becomes accessible again. Our idea of TEC is similar to the "Timer Proteins" previously proposed by Ref. [77], but in contrast, it does not require an active selection of feedforward protein activations to produce a specific conditional response. Recently, a different biochemical mechanism was proposed for time interval learning, which uses $Ca^{+2}$ oscillation and feedback loops for storing different time intervals information [78]. Unlike our mathematical models which explain the temporal profile of the conditional response after learning has been completed, their mathematical model focuses on the conditional learning process, which occurs during training in the Purkinje cell. However, their proposed underlying learning mechanism is fundamentally different from our biochemical mechanism in the sense that it does not require translocation of receptors at the synapse. Yet, another potential mechanism behind time-memory learning involves microtubules along with a mitogen-activated protein kinase (MAPK) pathway based on its general role in memory and learning processes [79]. This mechanism considers alteration in microtubules dynamics and their hexgonal lattice structure, which ultimately leads to storage of different time-duration memories. However, this does not directly explain how the Purkinje cell suppresses its tonic firing rate. All these three alternative hypotheses—involving microtubules, the $Ca^{+2}$ mechanism as well as our GIRK ion channel dependent suppression of the firing rate—solely consider molecular interactions within individual Purkinje cells and assume that mGluR$_7$ receptors initiate the conditional response.

As mentioned in the Introduction, the presence of mGluR$_7$ receptors on Purkinje cells is highly controversial. Previous studies by Johansson, Hesslow and coworkers [12, 13, 22] provide ample indirect evidence for the presence of mGluR$_7$ receptors in PC synapses, but a proper immunohistochemical characterization within synaptic terminals is still necessary to fully confirm its presence. Another option would be to reverse trace the chemical cascade involving the GIRK ion channel, as it has been shown to cause the suppression in tonic firing rate in Purkinje cells [22] and is expressed in Purkinje cell synapses [37]. As the release of glutamate from the parallel fiber initiates the conditional response, one can conclude that only class 2 and 3 of the mGluR receptor family can be responsible for the initiation of the conditional response [80]. This is because only class 1 mGluR receptors interact with G$_{q/11}$ type G-proteins [80], which activate the Phospholipase C (PLC) enzyme. PLC converts Phosphatidylinositol 4,5-bisphosphate (PIP$_2$) molecules into diacylglycerol (DAG) and inositol 1,4,5-trisphosphate (IP3) [81]. As PIP$_2$ is essential for the GIRK ion channel activation [54], class 1 of mGluR cannot initiate the conditional response as shown in [82]. Among classes 2 and 3 of the mGluR receptor family, only the mGluR$_7$ receptor is expressed by the Purkinje cell which is based on mRNA expression levels mentioned earlier [26, 27].

An alternative pathway involves strong feed-forward inhibition from the molecular layer interneurons to the Purkinje cell [43, 83]. However, in this scenario the mechanism behind the precise timing of the conditional response has not been established yet. Due to the fact that neurons are inherently noisy, it is also a possibility that multiple mechanisms both at the level of individual Purkinje cells and at the network level including interneurons are responsible for the robust expression of the conditional response. At the single cell level, GIRK ion channels control different features. Besides suppressing the tonic firing rate of the Purkinje cell, GIRK ion channels also happen to potentiate pf-PC synapses [84] potentially by increasing the glutamate release from parallel fibers. Such potentiation of pf-PC synapses favors the robust activation of mGluR$_7$ receptors in order to access the time memory stored at synapses. At the network level, inhibition from interneurons effectively reduces the total GIRK ion channel current required to suppress the tonic firing rate of the entire Purkinje cell. In other words, fewer Purkinje cell's synapses would be needed to initiate the conditional response. Thus, having multiple ways to suppress the firing rate increases the memory storage capacity of an individual Purkinje cell within the cerebellum.

As previously mentioned, in our model the time information of the conditional response is stored in the TECs found on individual synapses, implying that the substrate or the Engram of a time memory can reside at individual synapses, not in a cell or a cell assembly. This result is in line with the synaptogenic point of view of memory substrates [10], where single synapses play a large role in memory formation. In contrast, another point of view puts more emphasis on the intrinsic plasticity of a whole neuronal cell compared to the synaptic plasticity of individual synapses [85]. Intrinsic plasticity considers changes in the electrophysiological properties of the cell by changing the expression of Voltage-dependent Ca/K ion channels and many other kinds of ion channels, which are expressed by neurons and which decide neural firing rate as well as the sensitivity of the cell upon stimulation. However, neither points of view can fully account for the development of the conditional response in the Purkinje cell, since it neither involves the formation or elimination of pf-PC synapses [12, 13], nor LTD of pf-PC synapses [14] nor any change in the electrophysiological properties of the cell [12]. Thus, Purkinje cells show a novel form of synaptic plasticity and provide an example of monosynaptic memory encoding. In addition, considering this fact and that each Purkinje cell makes at least one synapse with up to 200,000 parallel fibers passing through the dendritic tree of the cell [86], the Purkinje cell might be considered as a multi-information storage device. Specifically, one might be able to encode a specific time interval by stimulating only a subset of parallel fibers

and encode another time interval by stimulating a separate subset of fibers. In this case, a specific time memory out of the whole set can be selectively retrieved when the respective set of parallel fibers becomes active upon stimulation, producing the conditional response for the previously encoded time interval.

One can get a rough estimate of the total number of unique time memories that can be stored by an individual Purkinje cell by taking the ratio of the collective hyperpolarization current produced by GIRK ion channels from all synapses and the minimum required hyperpolarization current in order to noticeably suppress the tonic firing rate of the cell. To determine the minimum hyperpolarization current, one can assume that its value is approximately equal to the net resurgent $Na^+$ current as the resurgent $Na^+$ ion channel has the capability to spontaneously generate rapid sequences of action potentials [58, 87]. In order to determine the collective hyperpolarization current, one needs to know the conductances of the GIRK ion channels and their respective densities on the synapses. Although there are experimentally measured values for single GIRK ion channel conductances [88, 89], no absolute density quantification of GIRK ion channels at Purkinje cell synapses has been done as far as we know. Only relative abundances of GIRK ion channels at Purkinje cell's dendritic spines are known [37]. Hence, it is currently not possible to determine the collective hyperpolarization current and, thus, the total number of unique time memories that can be stored by an individual Purkinje cell. This remains an interesting challenge for the future.

As an alternative approach, one could aim to establish experimentally that an individual Purkinje cell can indeed store at least two different time memories at separate sets of pf-PC synapses. As discussed above, stimulating separate sets of parallel fibers can in principle initiate different conditional responses. While this can be achieved by electrodes [12], it is challenging less so in terms of potential experimental protocols for conditional training [43] but rather due to difficulties in selecting specific fibers. An alternative could be to stimulate granule cells in the Granule layer of the Cerebellum [90] since parallel fibers are axonal branches of the granule cells. By stimulating a selected sub-population of granule cells and a specific branch of the climbing fiber, a subset of pf-PC synapses of an individual Purkinje cells can be trained for a specific ISI. Stimulating granule cells may appear as a drawback as they also excite other GABAergic interneurons, namely Golgi, stellate, and basket cells, which directly or indirectly can influence Purkinje cell firing activity [90]. However, their excitation did not appear to influence the conditional response profile of the Purkinje cell as shown experimentally [12]. Hence, stimulating subsets of granule cells experimentally—potentially using optogenetics [91]—might be a good way to test the capability of a Purkinje cell as a multi-information storage device in the future.

*Note added*: New support for our proposed biochemical mechanism for time-encoding memory formation comes from the observation that the $mGluR_1$ receptor is necessary for the learning process, while it is not for the activation of the conditional response [82]. Despite using a different experimental approach, it basically verifies point 3 listed in the section "Specific experimental options to test the proposed model". Specifically, the observation matches with our proposed mechanism since the latter assumes that the $mGluR_1$ receptor is responsible for the learning via facilitating trafficking of the $mGluR_7$ receptor to the synapses. Further experimental verification of our proposed mechanisms potentially along the lines outlined above remains an exciting challenge for the future.

## Supporting information

**S1 Text. Parameter values of the Purkinje cell model.**
(PDF)

**S2 Text. Hill's equation.**
(PDF)

**S3 Text. Details of the comprehensive mathematical model.**
(PDF)

**S1 Table. Values of kinetic parameters used in the comprehensive mathematical model.**
(PDF)

**S2 Table. (Initial) Concentration of various biomolecules for comprehensive mathematical model.**
(PDF)

**S3 Table. Parameter sensitivity analysis of the comprehensive mathematical model.**
(PDF)

**S4 Table. Parameters values for the modified G-protein subunit dynamics.**
(PDF)

## Acknowledgments

The authors thank Dr. Ray W. Turner at the University of Calgary for helpful discussions.

## Author Contributions

**Conceptualization:** Ayush Mandwal, Javier G. Orlandi, Christoph Simon, Jörn Davidsen.

**Data curation:** Ayush Mandwal.

**Formal analysis:** Ayush Mandwal, Javier G. Orlandi, Jörn Davidsen.

**Funding acquisition:** Jörn Davidsen.

**Investigation:** Ayush Mandwal.

**Methodology:** Ayush Mandwal, Javier G. Orlandi, Jörn Davidsen.

**Project administration:** Ayush Mandwal, Javier G. Orlandi, Jörn Davidsen.

**Resources:** Jörn Davidsen.

**Software:** Ayush Mandwal.

**Supervision:** Javier G. Orlandi, Jörn Davidsen.

**Validation:** Ayush Mandwal, Javier G. Orlandi, Jörn Davidsen.

**Visualization:** Ayush Mandwal, Javier G. Orlandi.

**Writing – original draft:** Ayush Mandwal, Javier G. Orlandi, Jörn Davidsen.

**Writing – review & editing:** Ayush Mandwal, Javier G. Orlandi, Christoph Simon, Jörn Davidsen.

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
