## [Decision Letter · Decision Letter 0]

8 Dec 2020

PONE-D-20-33032

A biochemical mechanism for time-encoding memory formation within individual synapses of Purkinje cells

PLOS ONE

Dear Dr. Mandwal,

Thank you for submitting your manuscript to PLOS ONE. After careful consideration, we feel that it has merit but does not fully meet PLOS ONE’s publication criteria as it currently stands. Therefore, we invite you to submit a revised version of the manuscript that addresses the points raised during the review process.

Given the delay in resubmission, although understandable in these times, your manuscript has been reviewed by two new reviewers. Both agree that the manuscript has merits, as did the previous reviewers. Nevertheless, both also identified a number of issues that prevent acceptation of the manuscript in its current format. As you will find, most comments can be addressed by textual changes, following the suggestions made by the reviewers. These changes should focus at further clarifying the applied methodology and the interpretation of the results in view of the existing literature.

On top of the required changes in the text, especially reviewer 1 also addresses a few points that may require further experiments. The expression of mGluR7 by Purkinje cells needs to be unequivocally addressed, either by providing appropriate literature references or by experimental means. The second and third remark could be addressed experimentally, or the authors have to explain clearly why they would prefer not to do so.

We look forward to receiving your revised manuscript.

Kind regards,

Laurens W. J. Bosman, Ph.D.

Academic Editor

PLOS ONE

Journal Requirements:

 "This work was financially supported by the Natural Sciences and Engineering

Research Council of Canada through a Discovery Grant to JD and the Eyes High Initiative of the University of Calgary (JD,JO).The funders had no role in study design, data collection and analysis, decision to publish, or preparation of the manuscript.".

i) Please provide an amended statement that declares *all* the funding or sources of support (whether external or internal to your organization) received during this study, as detailed online in our guide for authors at http://journals.plos.org/plosone/s/submit-now.  Please also include the statement “There was no additional external funding received for this study.” in your updated Funding Statement.

ii) Please include your amended Funding Statement within your cover letter. We will change the online submission form on your behalf.

Reviewers' comments:

Reviewer's Responses to Questions

**Comments to the Author**

1. Is the manuscript technically sound, and do the data support the conclusions?

Reviewer #1: No

Reviewer #2: Partly

2. Has the statistical analysis been performed appropriately and rigorously? 

Reviewer #1: Yes

Reviewer #2: N/A

3. Have the authors made all data underlying the findings in their manuscript fully available?

Reviewer #1: Yes

Reviewer #2: Yes

4. Is the manuscript presented in an intelligible fashion and written in standard English?

Reviewer #1: Yes

Reviewer #2: Yes

5. Review Comments to the Author

Reviewer #1: Authors propose a novel cellular mechanism to explain the timing of Purkinje cell (PC) simple spike pauses as reported during Pavlovian eyeblink conditioning. This PC intrinsic timer mechanism would be initiated by mGluR7 activation ends with opening/closing of GIRK channels. This idea is largely based on experimental work from the Hesslow group. I agree with the authors that the concept of a cell-intrinsic timing mechanism is very attractive, since indeed, solely synaptic plasticity (LTD/LTP) seems unable to explain the millisecond precise timing of simple spike pauses. For this reason, I consider the work presented by Mandwal et al. as very relevant. I don’t feel myself in the position to comment on the modelling aspect of this paper. However, I have four comment on the biological assumptions/ implications of this work.

Major comments:

1) The central assumption of this intrinsic timing mechanism is that Purkinje cells express mGluR7. However, the evidence supporting this assumption is very thin. They refer to the Philips 1998 paper, but the specificity of the antibodies presented in this study is controversial. As far as we know, also based on Mouse Allen Brain atlas, mGluR7 is expressed in many forebrain structures, but the expression on PCs is very minimal, if not present at all. Authors should address this topic and cannot simply state that “mGluR7 resides on Purkinje cell synapses”. In addition, we challenge the authors to prove that mGluR7 is indeed in expressed in PCs, either experimentally or by providing better references.

2) Regarding the CR onset: in their modeled data, the CR onset is always around 75 ms after CS onset (see figure 9). We know that that mice indeed have a relatively fixed latency to CR onset, no matter the ISI (see Chettith and Medina paper), but other species like rabbits and ferrets, seem to have a CR onset that shift with the ISI. That means: the longer the ISI, the later the CS onset.

3) The shape of the Purkinje cells suppression for a long ISI (black trace in figure 9) is not very typical. The suppression should stop at the onset of the expected US. Same for green and red trace. Compare shape of simple spike suppression with Hesslow’s data, for instance the 2007 Jirenhed paper.

4) Authors mention the minimal ISI possible to train an animal (±100 ms). Based on their model, what is the longest ISI possible to train a Purkinje cell?

Minor comment:

Pavlovian conditioning is introduced with the cliché idea of a bell as CS. We challenge the authors to go back to Pavlov’s work and find a reference to ‘the bell’. Most probably Pavlov never used a bell.

The flow of the paper is often messy, sometimes poorly written, and need another round of thorough editing. For a next submission, please provide the figures ordered properly.

Reviewer #2: We appreciated a lot the novel work of the authors in simulating underlying mechanisms of cerebellum-driven associative learning, as a result of biochemical processes mediating intrinsic changes of Purkinje cell synapses. More detailed comments have been submitted as a separate file.

6. PLOS authors have the option to publish the peer review history of their article (what does this mean?). If published, this will include your full peer review and any attached files.

Reviewer #1: No

Reviewer #2: No

---

## [Author Response · Author response to Decision Letter 0]

26 Jan 2021

Reviewer #1: Authors propose a novel cellular mechanism to explain the timing of Purkinje cell (PC) simple spike pauses as reported during Pavlovian eyeblink conditioning. This PC intrinsic timer mechanism would be initiated by mGluR7 activation ends with opening/closing of GIRK channels. This idea is largely based on experimental work from the Hesslow group. I agree with the authors that the concept of a cell-intrinsic timing mechanism is very attractive, since indeed, solely synaptic plasticity (LTD/LTP) seems unable to explain the millisecond precise timing of simple spike pauses. For this reason, I consider the work presented by Mandwal et al. as very relevant. I don’t feel myself in the position to comment on the modelling aspect of this paper. However, I have four comment on the biological assumptions/ implications of this work.

Major comments:

1) The central assumption of this intrinsic timing mechanism is that Purkinje cells express mGluR7. However, the evidence supporting this assumption is very thin. They refer to the Philips 1998 paper, but the specificity of the antibodies presented in this study is controversial. As far as we know, also based on Mouse Allen Brain atlas, mGluR7 is expressed in many forebrain structures, but the expression on PCs is very minimal, if not present at all. Authors should address this topic and cannot simply state that “mGluR7 resides on Purkinje cell synapses”. In addition, we challenge the authors to prove that mGluR7 is indeed in expressed in PCs, either experimentally or by providing better references.

Reply: We acknowledge the referee concerns regarding mGluR7 expression in PC, especially regarding the Phillips 1998 results. Although the body of literature regarding mGluR7 in the cerebellum is limited, we believe there is strong enough evidence to support our claims. [Ohishi 1995] reports “mGluR7 mRNA was expressed moderately in the Purkinje” and “The expression of mGluR7 was observed in most of the Purkinje cells”. Similarly, [Kinzie 1995] report “Purkinje cells express mGluR7 mRNA at all postnatal stages”, although expression was low on adults, but still significant. These authors mention as a possibility that “this may be due in part to low cell density in the adult Purkinje cell layer”. Additionally, the effects of MMPIP as an mGluR7 selective antagonist replicate the results of mGluR7 knockouts [Masugi-Tokita 2015]. And the use of MMPIP in Johansson’s work [Johansson 2015] shows its effect on blocking the conditional response in PC. Additionally, we are uncertain with respect to the referee’s claims regarding the mGluR7 expression levels on the Allen atlas. Although the volumetric expression of mGluR7 in the cerebellum is indeed low, the expression power within the PC bodies is indeed significant. In fact, it presents similar levels as in the hippocampus (see cover letter for corresponding figure), where high levels of mGluR7 expression have been extensively reported in the literature. We have added the above references to our revised manuscript together with a corresponding discussion.

2) Regarding the CR onset: in their modeled data, the CR onset is always around 75 ms after CS onset (see figure 9). We know that that mice indeed have a relatively fixed latency to CR onset, no matter the ISI (see Chettith and Medina paper), but other species like rabbits and ferrets, seem to have a CR onset that shift with the ISI. That means: the longer the ISI, the later the CS onset.

Reply: We thank the reviewer for bringing this to our attention. Our mathematical models can indeed achieve both kinds of behaviour: fixed latency to CR onset and shifting CR onset with the ISIs. The former arises when the dynamics of the 4 units of G-protein that are required to activate the GIRK ion channel are strongly correlated or dependent. The latter arises when the dynamics are independent. Mathematically, this is captured by the functional form of the gating function of the GIRK ion channel in eq.(2), which depends linearly on hGIRK in the former case and takes on a hGIRK4 form in the latter case, see the newly added discussion in our manuscript for further details. Here, the value hGIRK depends on the activity of the G-protein as per our proposed biochemical mechanism. As mentioned by the Reviewer, Chettih and Medina (2011) experimentally observed for mice that the delay in CR onset remains constant at about 100 ms for different ISIs, which is quite close to our model results of 75 ms in the former case, see Figure 8. Figure 8 together with the new Table 3 now also shows that in the former case our model can quantitatively reproduce the experimentally observed shifts in the onset observed for ferrets by Johansson et al. (2015). We would like to emphasize that the change in the CR onset is the only change arising from the different gating functions of the GIRK ion channel. 

3) The shape of the Purkinje cells suppression for a long ISI (black trace in figure 9) is not very typical. The suppression should stop at the onset of the expected US. Same for green and red trace. Compare shape of simple spike suppression with Hesslow’s data, for instance the 2007 Jirenhed paper.

Reply: In fig. 9, we show that the suppression stops at or near to the onset of expected US. If there seems to be some delay in the rise of the firing rate (e.g. black trace), it is simply due to fact that we estimated the instantaneous firing rate by the inverse of the backward interspike interval due to the deterministic character of our model. In contrast, Jirenhed et al. (2007) estimate the average firing rate by using a bin size of 10ms over 40 recordings of Purkinje cells. Because of the intrinsic stochasticity across the recordings, they could calculate the average firing rate based on number of spikes in the bin. We cannot do that as our mathematical model is deterministic. Because of this difference, we cannot see the rise in the firing rate as shown in fig. 9 until we get a spike and that’s why it appears that the suppression does not stop immediately at the onset of US, especially for the long ISI. In the revised manuscript, we discuss this more clearly and we also show the instantaneous firing rate centered on the corresponding interspike interval indicating more clearly that indeed suppression stops at the onset of the expected US, see the revised Fig. 8.

4) Authors mention the minimal ISI possible to train an animal (±100 ms). Based on their model, what is the longest ISI possible to train a Purkinje cell?

Reply: The longest ISI duration ever recorded is about 1000 ms [1] but it is not clear that ISI durations cannot be longer than 1000 ms. Our comprehensive mathematical model produces longer ISI durations for smaller values of both the rate-constant of the G-protein, kgp, and the concentration of the biomolecule PDE. In particular, it can produce an ISI = 1000 ms or larger values as long as we can choose kgp and [PDE] to be arbitrary small but positive. As there is no literature to suggest a minimal value of kgp, we cannot say with confidence what the physical upper limit of the ISI duration is.

[1] Smith, M. C. (1968). CS-US interval and US intensity in classical conditioning of the rabbit’s nictitating membrane response. Journal of Comparative and Physiological Psychology, 66(3, Pt.1), 679–687. https://doi.org/10.1037/h0026550

Minor comment:

Pavlovian conditioning is introduced with the cliché idea of a bell as CS. We challenge the authors to go back to Pavlov’s work and find a reference to ‘the bell’. Most probably Pavlov never used a bell.

Reply: We thank the referee for pointing this out, we have rewritten the associated part of the introduction to be more historically accurate.

The flow of the paper is often messy, sometimes poorly written, and need another round of thorough editing. For a next submission, please provide the figures ordered properly.

Reply: Based on the comments of both reviewers, we have revised the structure of the paper and improved the writing to increase its readability. We apologize for the incorrect order of the figures.

Reviewer #2: We appreciated a lot the novel work of the authors in simulating underlying mechanisms of cerebellum-driven associative learning, as a result of biochemical processes mediating intrinsic changes of Purkinje cell synapses. More detailed comments have been submitted as a separate file.

The authors describe a novel model of associative learning based on biochemical mechanisms driving Purkinje cell intrinsic synaptic changes. The work is interesting for both neuroscientists investigating the cerebellum and computational neuroscientists modelling cerebellum-driven behaviors. An extensive description of their model conceptualization and implementation is provided and conclusions are solid. However, some major changes would help to improve the manuscript significantly, better supporting the conclusions and strengthening the impact it could have on cerebellum learning modelling. Those are explained more in detail below.

Conceptual comments:

1) the authors should comment the applicability of their model to delay and trace conditioning. Is the model in the paper applicable to both cases? In case of delay conditioning, stimuli should co-terminate, but this does not seem the case of the described model where the CS could end after the US (e.g. the black bar in Figure 4 representing CS ends after the green bar representing the ISI. Or did I misunderstand the meaning of bars in the figure?).

Reply: No, you did not misunderstand the meaning of bars in Fig. 4. To provide more context, both delay and trace conditioning are two different forms of associative learning procedure and differ in their training protocols. It is true that in trace conditioning, both Hippocampus and Cerebellum are involved for the eye-blink conditioning learning [1], the former helps to transforms trace conditioning into delay conditioning. However, the behaviour of the Purkinje cell is the same in both trace and delay conditioning [2]. In particular, Halverson et al. (2018) showed that both trace and delay eyelid conditioning engage the same PCs and that the cerebellar cortex operates under similar rules for delay and trace eyelid conditioning. This implies that the conditioning behaviour of Purkinje cells is independent of the training protocol. Our proposed mathematical model describes the conditional behaviour observed in the Purkinje cell after training has completed. In this sense, our model is applicable to both cases of conditioning. We have added a corresponding discuss to our manuscript.

 [1] Christian, K. M. (2003). Neural Substrates of Eyeblink Conditioning: Acquisition and Retention. Learning & Memory, 10(6), 427–455. https://doi.org/10.1101/lm.59603

[2] Halverson, H. E., Khilkevich, A., & Mauk, M. D. (2018). Cerebellar Processing Common to Delay and Trace Eyelid Conditioning. The Journal of Neuroscience, 38(33), 7221–7236. https://doi.org/10.1523/jneurosci.0430-18.2018

2) The role of repeated CS-US paired trials is not addressed in the model, as all figures report single-trial analysis. Is your model able to capture this aspect of learning (e.g. start of learning, middle of learning)? Or only after-learning responses could be simulated?

Reply: The proposed mathematical models in our paper can completely capture the conditional response after training is completed such that indeed only after-learning responses can be stimulated. We have stated this now more clearly throughout the paper. However, we do provide a general description of the potential mechanism behind learning in the Biochemical description section. During training, the amount of mGluR7 receptors increases which increases the net conductance of GIRK ion channels. This aspect of learning can be modeled by a monotonic increase in the net conductance of GIRK ion channel. However, other aspects of learning experimentally observed during training – such as the increase of the ISI duration of the conditional response until it reaches its final value [1] – cannot be easily captured by our current mathematical model. Such a behavior would require the adjustment of kgp and [PDE] in the comprehensive model or tau3 and beta in the minimal model over time. Modeling this aspect from a biological perspective is highly nontrivial as it involves the translocation and protein complex formation of the various proteins involved in the proposed mechanism. Hence, it remains a challenge for the future. We have added a corresponding paragraph to the Discussion section. 

[1] Jirenhed, D.-A., Bengtsson, F., & Hesslow, G. (2007). Acquisition, Extinction, and Reacquisition of a Cerebellar Cortical Memory Trace. Journal of Neuroscience, 27(10), 2493–2502. https://doi.org/10.1523/jneurosci.4202-06.2007

3) Model parameters: all of them are derived from experiments in literature or from other literature modelling studies? None of them required any tuning/fitting procedure? This should be clarified.

Reply: As described now more clearly in the subsection “Parameter values” within “Materials and Methods” as well as in S3 Text and S1 Table, almost all of the parameters of the comprehensive model are experimentally constrained. While not all parameters of the comprehensive model have been directly measured, the ones that haven’t were set within the physical bounds based on the literature. As our sensitivity analysis of the comprehensive model (and similarly the bifurcation and stability analysis of the minimal model) shows, our overall findings are largely robust against variations in these parameters and therefore do not require specific tuning/fitting procedures. Only the values of [PDE] and kgp in the comprehensive model were chosen or tuned to match the specific ISIs.

4) Being able to reproduce more detailed mechanisms and fit models to experimental data is not always a good reason to build detailed models: a part from this (“Yet, some of the parameters in our minimal model are effective parameters and, hence, cannot be connected to experimentally accessible parameters and molecular interactions.”), could you discuss more reasons and application examples where the detailed model should be preferred with respect to the minimal one?

Reply: The detailed model was proposed to support our simplified model. In the previous round of reviews, reviewers raised concerns about the validity of the simplified model and its parameter values. That’s why we developed a detailed model based on biochemical interactions between various biomolecules. As mentioned by the reviewer, this allowed us to take advantage of the existing literature to determine or at least constrain the model parameters. In addition, as it captures the intrinsic biochemical details, the detailed model allows us to study the effect of blocking partially or completely a single or multiple biochemicals together such as PDE or PP1. However, the simplified model can only offer limited knockout features due to the reduced number of variables. We now state this more clearly in the subsection “Comprehensive mathematic model of proposed biochemical mechanism” of our revised paper. 

5) The role of feedforward inhibition from Molecular Layer Interneurons is mentioned in the Introduction but not addressed in the Discussion: this should also be commented as another mechanism contributing to associative learning as reported in [Ten Brinke et al., Cell Reports, 2015] and [Boele et al., Science Adv, 2018].

Reply: We thank the reviewer for pointing this out. We now discuss both papers in the Discussion section.

The paper needs an extensive reorganization to be more concise and clear:

1) The Materials and Methods section clearly describing the contribution of the authors with respect to state of the art reported in the Introduction (including the design of the biochemical model and the minimal model). The section “Biochemical description” could be summarized, for example using a table to report experimental evidence and corresponding hypotheses made by authors; alternatively, a block diagram integrating the flow diagram of Figure 2 could help to make this part more clear and concise.

Reply: We have largely followed the suggestions and restructured and revised the “Materials and Methods” section. In our biochemical description, we provide evidence and arguments for what could or couldn’t be a potential mechanism behind learning of the conditional response such that we feel a table form is not appropriate.

2) For each step/analysis presented in “Materials and Methods” (model conceptualization, model implementation in minimal and full versions, stability and sensitivity analysis methods, parametrization of the model, simulation and simulation analysis), the Results should be properly described in a corresponding section within “Results”, avoiding duplication in different parts of the paper.

Reply: We have restructured the “Materials and Methods” section as suggested and we have removed duplications in the Results section and throughout as much as possible.

3) some sections (e.g. Hill’s model, tables with parameter values, kinetic equations of the comprehensive model – leaving only one example one in the main text) could go to Supplementary Material.

Reply: As suggested, we have moved some of the content into the Supplementary Material. 

4) Figure 2 and 3 are redundant.

Reply: We have removed fig 2.

5) Table 2 should be summarized and being complementary to the text, while some concepts are repeated.

Reply: We have removed all repetitions in the text such that now Table 1 (formerly Table 2) provides a concise description of the various terms of the minimal model.

6) The stability analysis should be more concise (a summary table could help)

Reply: Given the nature of the stability analysis, we feel that a summary table is not appropriate but we have shortened the presentation of the analysis as much as possible.

7) Figures reporting the same results for the minimal and full model (e.g. 4 and 12) could be merged together in single figures with multiple panels, reorganizing the results so that the observations common to both models are presented together.

Reply: We have followed the suggestion and combined the discussion of the minimal and full model in the Results section as well.

8) Model comparison and model limitations should go in Results or Discussion sections.

Reply: We have moved both to the Discussion section.

Clarification comments:

1) Figure 4: legend covers plot; green bar not explained

Reply: We thank the reviewer for pointing this out. The issue is now fixed in both Fig. 4 and Fig. 9.

2) Figure 6: where is the impact of γ values?

Reply: The impact is captured by the two different panels in that figure (now Fig. 5) that show the behavior for different values of gamma. We have added a corresponding statement on the impact of gamma to the figure captions. Note that the impact of gamma is discussed in more detail in the robustness and stability section.

3) How was Equ. 1 derived? From literature?

Reply: Eq.1 (now Eq. 2) is a simplified equation for the gating of an ion channel. GIRK ion channels open via binding of 4 units of Gβγ. Each of these subunits has their own interaction with the active receptor (G-protein coupled receptor) and GAPs (GTPase activating proteins). To simply the complexity involved in the activation of a GIRK ion channel, we used a standard ion channel gating equation as used in [1] which has a gating parameter hGIRK, a voltage dependence function VGIRK(V) and a parameter for the channel’s net conductance gGIRK. As the GIRK ion channel binds 4 Gβγ, we used a functional form of h4GIRK. The voltage dependence of the GIRK ion channel is based on the literature, while the conductance value has not been established experimentally for Purkinje cells yet and it is chosen to match the experimentally observed conditional response profiles. All these aspects are now more clearly discussed in the subsection “Purkinje cell model”.

[1] Izhikevich, E. M. (2010). Dynamical Systems in Neuroscience: The Geometry of Excitability and Bursting (Computational Neuroscience) by Izhikevich, Eugene M. (2010) Paperback (Computational Neuroscience Series) (24383rd ed.). The MIT Press.

4) Which software was used for simulations? Is code available? If yes, link to repository should be added in the manuscript (data availability is not applicable, but code should be made available)

Reply: We used the publicly available odeint module in the scipy library of python to integrate all ordinary differential equations. We have added a corresponding subsection “Numerical simulations” with all details to our paper.

5) Figure 13: the short stimulus is not tested? (As in Figure 7; while in Figure 7 ISI = 400ms is not tested). Those two figures should be completed. Have those modelling results been validated against data (e.g. same % decrease of Purkinje cell firing rate obtained in the model with respect to experiments?)

Reply: Yes, it was tested. In particular, the effects of long and short CS are already shown in fig. 4 and fig.12 (now combined in fig. 4), which directly applies to fig.7 and fig.13 (now combined in fig. 6). In that sense adding more panels seems redundant to us. Regarding the validation against data, fig. 8 allows us to do exactly that. For example, the Purkinje cell firing rate in experiments shows a drop roughly between 10 and 30% [Johannsen et al., Cell Reports, 2015], while both our models show a drop in firing rate between 20 and 30% for the chosen value of gGIRK. As we now discuss more clearly in the manuscript, a number of other aspects of our models are also consistent with experimental data (see in particular our reply to Reviewer 1’s points 2 and 3).

Minor comments:

l. 26-31: it should be specified that the Conditioned Response is generated after repeated paired

presentations of the two stimuli

l. 34: the projection to cerebellar nuclei should be mentioned in describing how PC activity change controls ocular muscles.

l. 122: “the ” before learning should be removed

l. 351: “=λ” ?

l. 642: acronym TEC had already been defined

l. 715: “Note added” looks like a draft

A table with acronyms would help

Reply: We thank the reviewer for pointing these things out, we have fixed all issues.

---

## [Decision Letter · Decision Letter 1]

23 Feb 2021

PONE-D-20-33032R1

A biochemical mechanism for time-encoding memory formation within individual synapses of Purkinje cells

PLOS ONE

Dear Dr. Mandwal,

Thank you for submitting your manuscript to PLOS ONE. After careful consideration, we feel that it has merit but does not fully meet PLOS ONE’s publication criteria as it currently stands. Therefore, we invite you to submit a revised version of the manuscript that addresses the points raised during the review process.

We look forward to receiving your revised manuscript.

Kind regards,

Laurens W. J. Bosman, Ph.D.

Academic Editor

PLOS ONE

Additional Editor Comments (if provided):

Dear Dr. Mandwal,

First of all, I would like to thank you for the revised version of your manuscript. This version has been evaluated by the same two reviewers as during the previous round. As you will notice, both reviewers agree that the new version is a clear improvement.

Both reviewers offer a number of minor suggestions for improving the text. In my opinion, these are issues that can be relatively easily solved in the text.

However, reviewer 1 is not satisfied with the evidence presented regarding the presence of mGluR7 in Purkinje cell synapses. mGluR7 plays a central role in the current manuscript (see for instance lines 101-103: "Thus, we start from the assumption that Purkinje cells express mGluR7 and that the mGluR7 receptors indeed activate the conditional response behaviour in the Purkinje cell."). Consequently, the presence of mGluR7 is crucial for the argumentation presented in this manuscript. Reviewer 1 indicates that, in their eyes, mGluR7 protein expression in Purkinje cells is disputed, and may be restricted to terminals in the cerebellar nuclei. The authors point a.o. to RNA-expression (Allen Brain Atlas data), but this does not address the subcellular expression pattern.

I hope you will understand that this poses a problem for me as editor. Given the general positive feeling of the reviewers and myself, and given the substantial effort that you delivered in improving the manuscript, I would feel sorry to reject this manuscript. On the other hand, the current controversy regarding mGluR7 expression by Purkinje cells is too central to the manuscript to ignore. I see two solutions: either the authors take up the challenge of reviewer 1 and demonstrate convincingly that mGluR7 protein is present at the subcellular sites as assumed in the manuscript, or the authors acknowledge that there are substantial challenges to the current assumption. If the authors have a better idea, I will be open to that solution as well. Following option 1, the issue of putative aspecific staining should be addressed, following option 2, it should be made clear what the consequences are for the (other) conclusions of this study.

Kind regards,

Laurens Bosman

Reviewers' comments:

Reviewer's Responses to Questions

**Comments to the Author**

1. If the authors have adequately addressed your comments raised in a previous round of review and you feel that this manuscript is now acceptable for publication, you may indicate that here to bypass the “Comments to the Author” section, enter your conflict of interest statement in the “Confidential to Editor” section, and submit your "Accept" recommendation.

Reviewer #1: (No Response)

Reviewer #2: (No Response)

2. Is the manuscript technically sound, and do the data support the conclusions?

Reviewer #1: No

Reviewer #2: Yes

3. Has the statistical analysis been performed appropriately and rigorously? 

Reviewer #1: Yes

Reviewer #2: N/A

4. Have the authors made all data underlying the findings in their manuscript fully available?

Reviewer #1: Yes

Reviewer #2: Yes

5. Is the manuscript presented in an intelligible fashion and written in standard English?

Reviewer #1: Yes

Reviewer #2: Yes

6. Review Comments to the Author

Reviewer #1: 1) mGluR7 expression on PC:

The authors have added a few sentences to the material and methods, but do not seem to realize the controversy around the statement that ‘mGluR7 resides on Purkinje cells”. The authors have added:

"Although the body of literature regarding mGluR7 in the cerebellum is limited and despite some controversy [29], there is significant direct evidence that Purkinje cells do express mGluR7 receptors on their entire cell body and dendritic branches [30, 32]. More importantly, the effects of 6-(4-Methoxyphenyl)-5-methyl-3-(4-pyridinyl)isoxazolo[4,5-c]pyridin-4(5H)-one hydrochloride (MMPIP) as an mGluR7 selective antagonist replicate the results of mGluR7 knockouts [33], while MMPIP also has an effect on blocking the conditional response in Purkinje cells [13]. Thus, we start from the assumption that Purkinje cells express mGluR7 and that the mGluR7 receptors indeed activate the conditional response behaviour in the Purkinje cell."

They refer to the following papers:

Ref 29: Kinoshita et al., 1998

This is absolutely the best (and most recent) study that addresses mGluR7 expression on PC. This study cannot be ignored by simply saying “some controversy”. The Kinoshita et al., 1998 study is the only study that makes use of proper mGluR7 KO controls. All the other studies lack proper controls (i.e. mGluR7 KO). Kinoshita et al., 1998 reports: “Even in the mGluR7-deficient mouse, however, weak immunostaining was occasionally seen in the cytoplasm of neuronal cell bodies, especially in large neurons including Purkinje cells and motor neurons, as well as in Bergmann’s glia in the cerebellar cortex. Therefore, weak immunostaining which was occasionally observed in neuronal cell bodies and Bergmann’s glia in the rat and wild-type mouse was not considered to represent mGluR7-LI. Thus, neither mGluR7a- nor mGluR7b-LI was found in neuronal cell bodies in the present study."

Instead, the Kinoshita 1998 conclusively points towards a presynaptic localization of mGluR7b on PC axon terminals in the cerebellar nuclei (Fig 14) and “Purkinje cells, though not equipped with mGluR7a, might express GluR7b in their axon terminals of recurrent axon collaterals, as well as in those of projection fibers to the deep cerebellar nuclei and lateral vestibular nucleus”

Ref 30: Philips et al 1998:

This study lacks controls.

Ref 32: Ohishi et al, 1995:

Similar to 30: lack of controls, this is the on of the reason why the same group repeated this study and published it as Kinoshita et al., 1998.

Ref 33: Kinzie et al., 1995:

No controls and the presented data lacks detail. Without showing data, they report “In the adult, mGluR7 mRNA expression all but disappears in granule cells, while mGluR7 mRNA expression in Purkinje cells is maintained.

Thus, the central assumption of mGluR7 residing on PC dendritic spines (fig1) is most likely wrong. Again, I challenge the authors to provide experimental data supporting mGluR7 expression in PC. Otherwise, the authors have to acknowledge the controversy in a more prominent place in the manuscript (introduction!), without simply saying ‘some controversy” while they refer to the best work done on PC mGluR7 expression.

2) CR onset

No further comments.

3) PC suppression

No further comments, well explained.

4) ISI duration

No further comments, thanks.

Further comments:

Include: Yousefzadeh et al 2020, Internal clocks, mGluR7 and microtubules.

Reviewer #2: We thank the Authors for having addressed the comments raised in the first review round. The manuscript is significantly improved. Some additional points that should still be addressed are reported in the attached file.

7. PLOS authors have the option to publish the peer review history of their article (what does this mean?). If published, this will include your full peer review and any attached files.

Reviewer #1: No

Reviewer #2: No

---

## [Author Response · Author response to Decision Letter 1]

13 Apr 2021

Reviewer #1: 1) mGluR7 expression on PC:

The authors have added a few sentences to the material and methods, but do not seem to realize the controversy around the statement that ‘mGluR7 resides on Purkinje cells”. The authors have added:

"Although the body of literature regarding mGluR7 in the cerebellum is limited and despite some controversy [29], there is significant direct evidence that Purkinje cells do express mGluR7 receptors on their entire cell body and dendritic branches [30, 32]. More importantly, the effects of 6-(4-Methoxyphenyl)-5-methyl-3-(4-pyridinyl)isoxazolo[4,5-c]pyridin-4(5H)-one hydrochloride (MMPIP) as an mGluR7 selective antagonist replicate the results of mGluR7 knockouts [33], while MMPIP also has an effect on blocking the conditional response in Purkinje cells [13]. Thus, we start from the assumption that Purkinje cells express mGluR7 and that the mGluR7 receptors indeed activate the conditional response behaviour in the Purkinje cell."

They refer to the following papers:

Ref 29: Kinoshita et al., 1998

This is absolutely the best (and most recent) study that addresses mGluR7 expression on PC. This study cannot be ignored by simply saying “some controversy”. The Kinoshita et al., 1998 study is the only study that makes use of proper mGluR7 KO controls. All the other studies lack proper controls (i.e. mGluR7 KO). Kinoshita et al., 1998 reports: “Even in the mGluR7-deficient mouse, however, weak immunostaining was occasionally seen in the cytoplasm of neuronal cell bodies, especially in large neurons including Purkinje cells and motor neurons, as well as in Bergmann’s glia in the cerebellar cortex. Therefore, weak immunostaining which was occasionally observed in neuronal cell bodies and Bergmann’s glia in the rat and wild-type mouse was not considered to represent mGluR7-LI. Thus, neither mGluR7a- nor mGluR7b-LI was found in neuronal cell bodies in the present study."

Instead, the Kinoshita 1998 conclusively points towards a presynaptic localization of mGluR7b on PC axon terminals in the cerebellar nuclei (Fig 14) and “Purkinje cells, though not equipped with mGluR7a, might express GluR7b in their axon terminals of recurrent axon collaterals, as well as in those of projection fibers to the deep cerebellar nuclei and lateral vestibular nucleus”

Ref 30: Philips et al 1998:

This study lacks controls.

Ref 32: Ohishi et al, 1995:

Similar to 30: lack of controls, this is the on of the reason why the same group repeated this study and published it as Kinoshita et al., 1998.

Ref 33: Kinzie et al., 1995:

No controls and the presented data lacks detail. Without showing data, they report “In the adult, mGluR7 mRNA expression all but disappears in granule cells, while mGluR7 mRNA expression in Purkinje cells is maintained.

Thus, the central assumption of mGluR7 residing on PC dendritic spines (fig1) is most likely wrong. Again, I challenge the authors to provide experimental data supporting mGluR7 expression in PC. Otherwise, the authors have to acknowledge the controversy in a more prominent place in the manuscript (introduction!), without simply saying ‘some controversy” while they refer to the best work done on PC mGluR7 expression.

Reply: We thank the referee for clearly summarizing the histochemical evidence of mGluR7 expression in PC. We understand and acknowledge the concerns and have modified our paper throughout accordingly. This includes two new paragraphs we added to the introduction and discussion section. It is beyond the scope of this paper to provide experimental evidence of mGluR7 expression on the PC synapses. However, the activity-based and indirect findings presented by Johansson, Hesslow and co-workes are convincing enough, in our view, to warrant our study. In particular, the work by Johansson et al. published in 2015 in Cell Reports clearly shows that mGluR7 antagonists act postsynaptically in PCs. They already discuss other possibilities in their discussion section, and among others “Using a multibarrel electrode, we were able to inject very small amounts of antagonist only micrometers away from the Purkinje cell dendrites from which we recorded and the effects were immediate, making other targets very unlikely.”

2) CR onset

No further comments.

3) PC suppression

No further comments, well explained.

4) ISI duration

No further comments, thanks.

Further comments:

Include: Yousefzadeh et al 2020, Internal clocks, mGluR7 and microtubules.

Reply: Thank you for pointing out this reference. We now cite this work and discuss it in the Discussion section.

##########################################################################

Reviewer #2: We thank the Authors for having addressed the comments raised in the first review round. The manuscript is significantly improved. Some additional points that should still be addressed are reported in the attached file.

The authors significantly improved the readability of the paper and addressed the previous comments. In general, some additional effort is required to contextualize the paper and the modelling work within theories of conditioned response learning, and to present the impacts of the current results (besides suggesting some possible new experiments to test model predictions). Detailed comments are suggested below to further improve the manuscript.

l.106-121: this part on the alternative hypotheses should be summarized, going directly to the conclusion and citing the literature.

Reply: We have shortened and summarized this part as much as possible.

l.121-137: the correlation between the protein mechanisms and the need for 2 stimuli (one from PFs and the other from CFs), to learn the response is not clear to me from this paragraph; please, try to rephrase.

Reply: We have revised this paragraph to clarify the need for two stimuli.

The paragraph “Training with different ISI duration and time-encoding protein complexes” could be summarized more, going directly to the point and summarizing reasoning steps.

Reply: Given that the paragraph is already quite short, we believe that summarizing it even further will make some of the aspects unnecessarily challenging and or even inaccessible for the non-expert reader.

l.266-272: this paragraph introduces synapse potentiation/depression comments that should be better moved to the Discussion.

Reply: We have followed this suggestion.

l.305-307: this comment should go directly in the paragraph where you discuss this.

Reply: We have moved this comment to the preceding paragraph.

l.559: the reference of experimental study is missing; it should be clearly stated in each part which experimental studies you are referring to, in order to compare model results to measures.

Reply: Thank you for pointing this out. We have added the missing reference.

Fig. 3: labels of bottom panel should be changed. Please, also change colors for better clarity. From the figure it also looks like the v has the same trend between short and long ISI. How long are the short and long ISI values should be reported (in a simulation protocol paragraph of the methods, where the tests done should be described). How were the values chosen? Is the short ISI lower than 100ms?

Reply: We have followed the suggestion and revised both the labels for the tick marks and our color choice to improve readability. We would like to note that Fig. 3 shows the case of a single ISI (200 ms as indicated by the green vertical bars). What is varied between the different panels is the duration of the stimulus (indicated by the horizontal bars). The duration of the short stimulus (20 msec) was selected based on the Johansson et. al (2015) paper, while the long stimulus duration (>2.5 sec) is arbitrary and serves to show that the conditional response is indeed independent of stimulus duration. We have revised the captions to clarify this. For a comparison between different ISI’s (200 – 400ms), please see Figs. 7 and 8. Regarding the minimum and maximum ISI durations, both our models can produce ISI=100 msec and ISI=1000 msec, which are the extreme values observed experimentally [1,2]. This is simply because β or equivalently [PDE] can take on a wide range of values without fundamentally changing the behavior. This can be seen, for example, in the bifurcation plot (Fig.5) for δ=1.0 and γ=1.4. Larger values of β imply shorter ISI durations of the conditional response and vice versa.

[1] Smith, M. C. (1968). CS-US interval and US intensity in classical conditioning of the rabbit’s nictitating membrane response. Journal of Comparative and Physiological Psychology, 66(3, Pt.1), 679–687. https://doi.org/10.1037/h0026550

[2] Wetmore DZ, Jirenhed DA, Rasmussen A, Johansson F, Schnitzer MJ, Hesslow G. Bidirectional Plasticity of Purkinje Cells Matches Temporal Features of Learning. The Journal of Neuroscience. 2014;34:1731–1737. doi:10.1523/jneurosci.2883-13.2014.

Fig. 5: β = 6.7645, is the model so sensitive to β values (4 th decimal digits specified)

Reply: This specific beta value denotes a bifurcation point in our minimal model. It is common to specify bifurcation points to the resolution of the numerical algorithm used to determine them, which is four digits in our case here. Note that a bifurcation point is a point where the behaviour of a given system changes significantly, such that indeed the model behavior for smaller and larger values of beta is fundamentally different.

Fig. 8: it is not clear how MMPIP enters in the model

Reply: As discussed in the main text, MMPIP is a mGluR7 selective antagonist. As mGluR7 is necessary for G-protein activation, which in turn activates GIRK ion channels, the application of MMPIP indirectly affects the fraction of the total GIRK ion channels which will be active in the presence of a conditional stimulus. Hence, for simulating the effect of increasing doses of MMPIP reagent, we simply reduce the gGIRK conductance value. This is now more clearly reiterated in the captions of Fig. 8.

l.766: the PC spike suppression should be quantified (reporting e.g. pause duration or drop in firing rate), possibly comparing to experimental data from reference papers

Reply: We apologize for our carelessness. We mentioned it in the previous reply letter but missed to include it in the manuscript. We have now included this information.

Paragraph “Specific experimental options to test the proposed model” is too detailed and could be summarized in a couple of examples.

Reply: We feel that it is already quite short especially since it is intended to stimulate future experiments.

l.870: the success of the model should be based on the comparison against data, not on the comparison between minimal and comprehensive model.

Reply: We agree with the reviewer and have rephrased the statement accordingly.

l.952: this comment should be stressed more throughout the paper (also in the Introduction). You are focusing on one single neuron level mechanism, how this impacts in network behavior? It is probably the whole network giving rise to learning as a results of multiple mechanisms at different scales.

Reply: We have followed this suggestion and stress this now throughout the manuscript. In particular, we have significantly extended the corresponding part in the discussion section.

l.1013: is this part intended to start with “Note added”?

Reply: Yes.

It is not clear to me how the proposed model takes into account the role of the signal pathway from climbing fibers carrying the US.

Reply: Our proposed model does not take into account the role of the signal pathway from the climbing fiber carrying the US. In particular, our mathematical model focusses on the conditional response after training has been completed. We mentioned the role of the signal pathway from climbing fibers in the subsection “During training: Learning process”, which explains how the presence of paired stimuli of CS and US leads to learning of the conditional response.

Minor comments:

l.64: I would change “with” to “within” as the Conditioned Response is anticipatory with respect to the second stimulus.

Reply: Done.

l.75: “decrease in cAMP concentration, [cAMP]” would be better.

Reply: Done.

l.187: “mechanims” typo

Reply: Done.

l.189: AC?

Reply: AC is the acetyl cyclase enzyme as defined earlier.

Table 1: missing parenthesis in second raw

Reply: Done.

Fig. 4: x axis label? Legend could be one for the 4 panels and include also filled and empty dots meaning.

Reply: Fixed.

Fig. 9: legends could be merged

Reply: Done

---

## [Editor Report · Decision Letter 2]

22 Apr 2021

A biochemical mechanism for time-encoding memory formation within individual synapses of Purkinje cells

PONE-D-20-33032R2

Dear Dr. Mandwal,

We’re pleased to inform you that your manuscript has been judged scientifically suitable for publication and will be formally accepted for publication once it meets all outstanding technical requirements.

Kind regards,

Laurens W. J. Bosman, Ph.D.

Academic Editor

PLOS ONE

Additional Editor Comments (optional):

I would like to thank the authors for providing a new version. I think the reviewers' comments are sufficiently addressed.
---

## [Editor Report · Acceptance letter]

26 Apr 2021

PONE-D-20-33032R2 

A biochemical mechanism for time-encoding memory formation within individual synapses of Purkinje cells 

Dear Dr. Mandwal:

I'm pleased to inform you that your manuscript has been deemed suitable for publication in PLOS ONE. Congratulations! Your manuscript is now with our production department. 

Kind regards, 

on behalf of

Dr. Laurens W. J. Bosman 

Academic Editor

PLOS ONE